



# Remote sensing of aerosols with small satellites in formation flight

Kirk Knobelspiesse[1] and Sreeja Nag[2,3]

[1]NASA Goddard Space Flight Center, Greenbelt, MD, USA
[2]Bay Area Environmental Research Institute, Petaluma, CA, USA
[3]NASA Ames Research Center, Moffett Field, CA, USA

*Correspondence to:* Kirk Knobelspiesse (kirk.knobelspiesse@nasa.gov)

**Abstract.** Determination of aerosol optical properties with orbital passive remote sensing is a difficult task, as observations often have limited information. Multi-angle instruments, such as the Multi-angle Imaging SpectroRadiometer (MISR) and the POlarization and Directionality of the Earth's Reflectances (POLDER), seek to address this by making information rich multi-angle observations, which can be used to better retrieve aerosol optical properties. The paradigm for such instruments is

that each angle view is made from one platform, with, for example, a gimbaled sensor or multiple fixed view angle sensors. This restricts the observing geometry to a plane within the scene Bidirectional Reflectance Distribution Function ($BRDF$) observed at the top of the atmosphere (TOA). New technological developments, however, support sensors on small satellites flying in formation, which could be a beneficial alternative. Such sensors may have only one viewing direction each, but the agility of small satellites allows one to control this direction and change it over time. When such agile satellites are flown in

formation and their sensors pointed to the same location at approximately the same time, they could sample a distributed set of geometries within the scene $BRDF$. In other words, observations from multiple satellites can take a variety view zenith and azimuth angles, and are not restricted to one azimuth plane as is the case with a single multi-angle instrument. It is not known, however, if this is as potentially capable as a multi-angle platform for the purposes of aerosol remote sensing. Using a systems engineering tool coupled with an information content analysis technique, we investigate the feasibility of such an

approach for the remote sensing of aerosols. These tools test the mean results of all geometries encountered in an orbit. We find that small satellites in formation are equally capable as multi-angle platforms for aerosol remote sensing, as long as their calibration accuracies and measurement uncertainties are equivalent. As long as the viewing geometries are dispersed throughout the $BRDF$, it appears the quantity of view angles determines the information content of the observations, not the specific observation geometry. Given the smoothly varying nature of $BRDF$'s observed at the TOA, this is reasonable,

and supports the viability of aerosol remote sensing with small satellites flying in formation. The incremental improvement in information content that we found with number of view angles also supports the concept of a resilient mission comprised of multiple satellites that are continuously replaced as they age or fail.

## 1 Introduction

Atmospheric aerosols play a potentially significant role in the global climate, both through direct scattering and absorption

of solar radiation, and indirectly by modifying clouds and local meteorology. Additionally, aerosols contribute the largest





overall net radiative forcing uncertainty (IPCC (2013)), due in part to insufficiently accurate and complete observations on a global scale (Mishchenko et al. (2004)). This is because most aerosol remote sensing is underdetermined, meaning there is less information contained in the observations than necessary to accurately extract the necessary aerosol descriptive parameters. The aerosol remote sensing community is therefore developing instruments that maximize "information content," by observing

a scene at multiple wavelengths, viewing angles, and polarimetric states (Kokhanovsky et al. (2015)).

Notable examples of instruments that make use of multi-angle observations include the Multi-angle Imaging SpectroRadiometer (MISR) and the POlarization and Directionality of the Earth's Reflectances (POLDER). MISR, launched on the NASA Terra spacecraft in 1999, observes in four spectral bands and nine view angles spread in the flight track direction (Diner et al. (1998); Kahn et al. (2005)). As of 2016, MISR is still operational, and has been collecting data for more than sixteen

years. Three versions of POLDER have collected data, the most recent and longest lived on the French CNES (Centre National d'Etudes Spatiales) PARASOL (Polarization and Anisotropy of Reflectance for Atmospheric Sciences coupled with Observations from a Lidar) spacecraft, launched in 2004 and removed from orbit in 2013. POLDER observed a scene with up to sixteen views in the along track direction, with nine spectral channels at visible and near-infrared wavelengths. Three of those channels were also sensitive to linear polarization (Fougnie et al. (2007); Hasekamp et al. (2011); Tanré et al. (2011); Dubovik

et al. (2011)). The Clouds and Earth Radiant Energy System (CERES) instruments have the capability to scan in an arbitrary solar-sensor azimuth plane, although such systems do not collect multiple angle views of the observed location at the same time (Smith et al. (2011); Wielicki et al. (1996)).

The instruments described above are what we call 'multi-angle' platform instruments, since all measurements are made from one instrument. New and rapidly developing technology has created the possibility that several individual instruments

can make a multi-angle observation in an entirely different manner. We consider formations of single view angle instruments in orbit, coordinated to observe the same point simultaneously. This approach may be advantageous for a variety of engineering, cost, or operational reasons. A formation of small satellites can make multi-spectral measurements of a ground spot at multiple angles simultaneously as they pass overhead using narrow field of view instruments in controlled formation flight (Nag et al. (2016a, b)). Fig. 1 shows a graphic for a multiple satellite case, where the relative positions of the satellites do not need to

be tightly controlled, but their relative attitudes do. Our proposed concept is aimed at improving only the angular coverage of measurements because the images collected by the satellites are expected to overlap. The spatial and temporal coverage of the formation will depend on the swath of any single sensor, and can be improved by flying a constellation of such formations

Aerosol optical properties are determined from an orbital measurement, $y$, that is a vector containing reflectances for various wavelengths, polarization states, and geometries. Such a vector represents a sample of the top of atmosphere (TOA) bidirec-

tional reflectance distribution function ($BRDF$). As defined in Nicodemus et al. (1977), the $BRDF$ is the geometrically and spectrally resolved ratio of scattered to incident light,

$$BRDF(\theta_s, \theta_v, \phi_s, \phi_v, \lambda) = \frac{dL(\theta_s, \theta_v, \phi_s, \phi_v, \lambda)}{dE(\theta_s, \phi, \lambda)} [sr^{-1}] \tag{1}$$





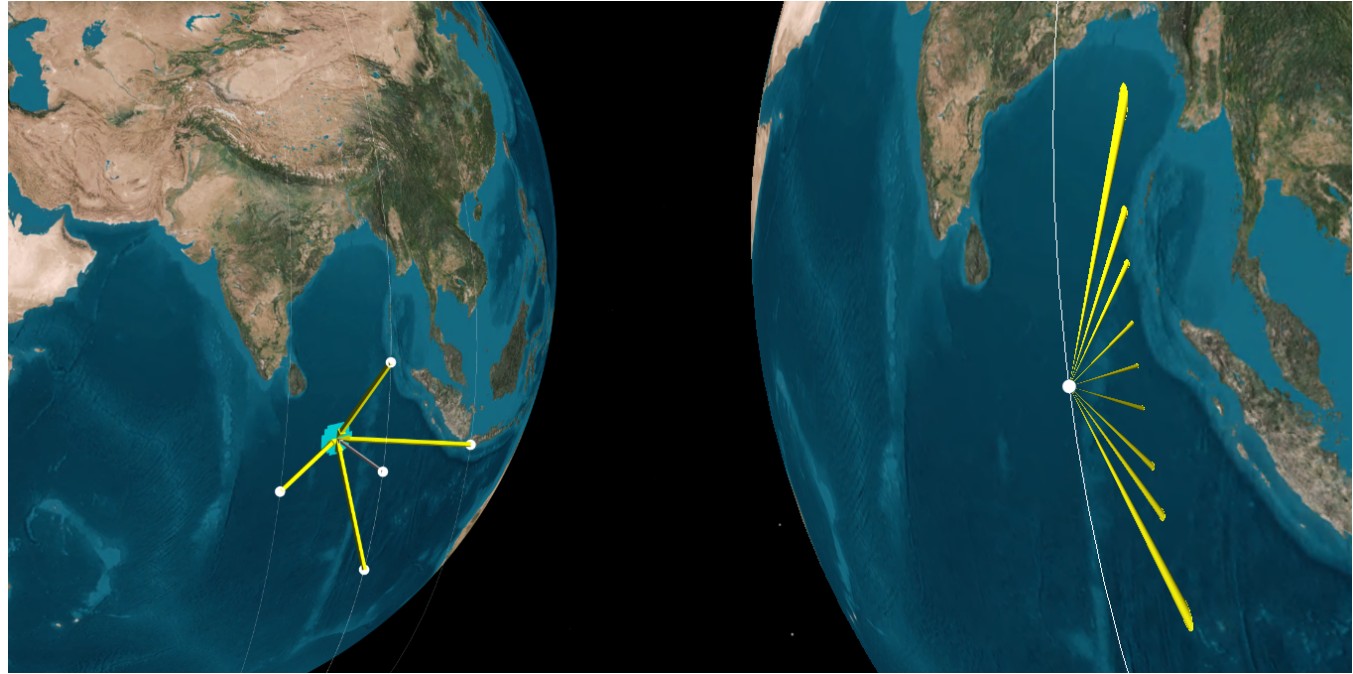

**Figure 1.** Illustration of the observation geometry of five, single view angle, satellites flying in formation (left) and a nine view multi-angle single satellite (right). The satellites in formation flight simultaneously observe the same ground spot at five different zenith and azimuth angles. The relative positions of these satellites do not need to be tightly controlled, but their relative attitudes do. The multi-angle satellite observes in the along track direction, so that a ground spot is observed from nine angles as the satellite passes overhead. These observations occur within the same azimuth plane at various zenith angles.

where $L$ is the radiance in units of $[W m^{-2} sr^{-1}]$ and $E$ is the irradiance in units of $[W m^{-2}]$. The $BRDF$ is a function of solar zenith angle, $\theta_s$, view zenith angle, $\theta_v$, solar azimuth angle, $\phi_s$, view azimuth angle $\phi_v$, and wavelength, $\lambda$. Note that $L$ and $E$ could contain vectors describing polarization state, in which case the above equation would represent the Bidirectional Polarization Distribution Function ($BPDF$) (Nadal and Breon (1999)). For the earth, $BRDF$ is typically symmetric about the solar azimuth angle, so that $\phi_s$ and $\phi_v$ can be condensed to $\phi = \phi_s - \phi_v$ (Knobelspiesse et al. (2008)), which was what was

5  used here. The algorithm used to determine the optimal formation flight architectures, which assessed their ability to determine $BRDF$ (Nag et al. (2015)) or any $BRDF$ dependent product (Nag et al. (2016b)), did take the asymmetric azimuth nature into account. In practice, observations are often expressed as a unitless quantity (reflectance, see section 3.3) that has been integrated over solar geometries and represents a finite view solid angle.

The TOA $BRDF$ or $BPDF$ depend upon interactions between incoming solar radiation and the gases, aerosols, clouds
10  and surfaces that comprise an earth scene. They therefore can contain information about the optical properties and quantities of these constituents. A generalized way to retrieve these values is to compare the measurements, $y$, to the computed result of a radiative transfer (RT) model simulation. Such models compute a $BRDF$, which is sampled to simulate measurement





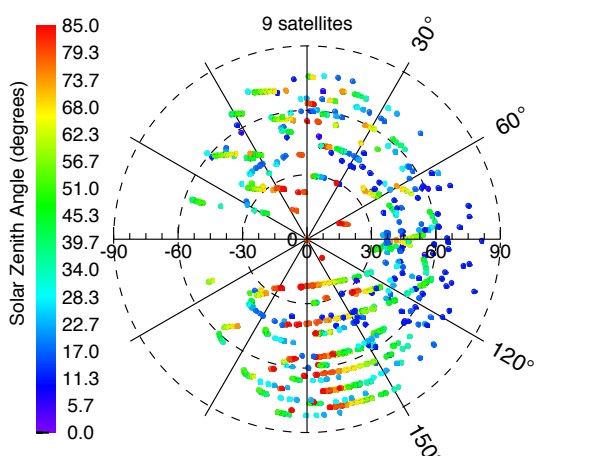
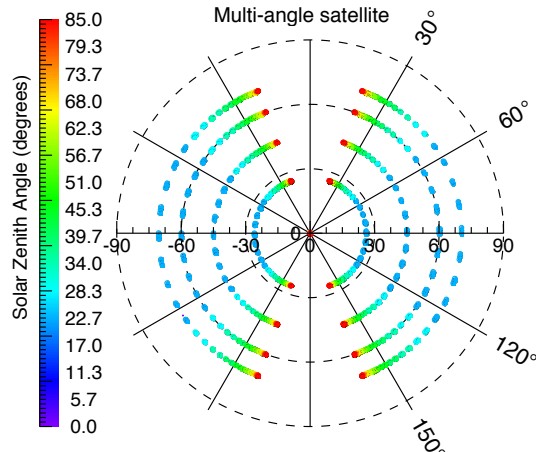

**Figure 2.** Simulated viewing geometries in one day for nine single view satellites in formation flight (left) and a multi-angle satellite with nine views in the along track direction (right). View zenith angle ($\theta_v$), is indicated in the radial coordinate dimension, while the relative solar - view angle azimuth ($\phi = \phi_s - \phi_v$) is in the angular dimension. Solar zenith angle ($\theta_s$) is expressed by color. We limited to $\theta_s < 85°$, 106 and 119 scenes for the nine satellite and multi-angle satellite configurations met this criteria in our tests, respectively.

vector $y'$ given a set of descriptive geophysical scene parameters, $x$. $y$ and $y'$ are then compared, and $x$ iteratively adjusted (by a variety of methods) until the closest match can be found. The ability to successfully converge to a solution depends on measurement system characteristics, RT model fidelity, and other factors. In this study, we are concerned with the impact that measurement characteristics, specifically observation geometry, have on the ability to accurately determine the portion of $x$ representing aerosols.

Fig. 2 illustrates the $BRDF$ sampling differences between a formation of instruments and a multi-angle instrument. Results were generated by simulations of orbit characteristics described in section 2.1, with roughly one hundred observations each for a formation of nine satellites (left) and a multi-angle instrument with a geometry similar to that of the MISR instrument (right). The multi-angle satellite observes the $BRDF$ or $BPDF$ in a much more ordered manner than the formation of nine satellites: observations are made at a fixed $\theta_v$, and solar zenith angles covary with $\phi$. Observations are not made in the solar principal plane

(where $\phi = 0°$) except at nadir ($\theta_v = 0°$). The nine satellite configuration, however, is far less uniform. Many observations are made in the solar principal plane (although often with high $\theta_s$). The multi-angle instrument thus has a measurement vector, $y$, that is much more uniformly ordered than that of the formation of single view instruments.

The goal of this paper it to examine these differences and determine if there are advantages (or disadvantages) of using formations of multiple satellites with single but adjustable view, compared to multi-angle instruments. Section 2.1 describes

the systems engineering model used to select the satellite formation characteristics and observation geometries expressed in Fig. 2. Section 2.2 describes the information content assessment technique, which uses instrument characteristics and RT



model simulations to predict the uncertainty in the retrieved $x$. Section 3 provides details on the characteristics of the systems engineering models, RT model, and information content assessment. Section 4 contains the results of this assessment, while section 5 concludes.

## 2 Background

An architecture is defined as a unique combination of design variables such as number of satellites, their orbit parameters, the spectrometer or polarimeter payload's field of view, pixel size, number of spectral bands, spectral resolution, communication bands for downlink, etc. The methodology employed to assess the optimal architectures and validate their aerosol retrieval capabilities couples systems engineering and information content analysis, a method of science performance evaluation. A trade-space of architectures can be analyzed by varying the design variables in the systems engineering model and assessing its effect on science products using information content assessment, as shown in Fig. 3. The left hand box generates multiple architectures by permuting different values of the design variables, sizes them to check their technical feasibility and costs them relative to one another. The systems engineering model can be simulated over any time horizon and divided into appropriate time steps. The right hand box evaluates the information content that can be retrieved from the angular spread of measurements, at every instant of time, for every architecture. We perform this assessment using Bayesian statistical techniques that connect simulated scenes to the potential geophysical parameter retrieval ability of a selected architecture. This assessment is performed for a variety of types of scenes, so that the aggregate result is more representative of global conditions.

### 2.1 Systems engineering model

In the last few years, several small satellite constellations with atmospheric science sensors have successfully flown (e.g., the Cyclone Global Navigation Satellite System or CYGNSS, Ruf et al. (2012)) or have been funded for imminent flight (e.g. TROPICS, Blackwell (2015)). While such systems demonstrate capability to house science payloads, the satellites in these constellations do not need to coordinate their measurement operations, and their attitude is fixed in local space. Our proposed formation requires that all satellites point to the same target at approximately the same time, which needs agility and consistent attitude control. Simulation studies have demonstrated that it is possible to maintain the orbits and orientation of small satellites in such a formation, using commercially available propulsion and control systems (Nag et al. (2016b)).

As described in previous literature (Nag et al. (2016a)), a modular systems engineering model is capable of simulating hundreds of small satellite formation flight architectures, constrained by current small spacecraft capabilities such as launch availability, propulsive correction capability and commercial attitude control as well as by $BRDF$ measurement requirements such as medium spatial resolution and full hemispherical angular spread. Such a model automatically balances technical trades between conflicting variables such as required ground pixel size and control stability, or required pixel size and off-pointing angles, or number of orbital planes and off-track angles. Therefore, the formation flight architectures generated by the model are optimized to ensure they are technically feasible. The outputs corresponding from each architecture are (among others), the angular spread of measurements on the ground at any given simulation instant, where the number of measurements will



be equal to the number of satellites (Fig. 1). Fig. 3 summarizes the coupling between the systems engineering model, which generates spacecraft formation architectures, and the science evaluation model, which assesses the information content within the angular and temporal measurements that the architectures are capable of making. The coupling may be an iterative one where science performance errors are used to inform better engineering design.

This paper focuses on only those design variables in the systems engineering model that pertain to orbital design and payload pointing strategies of a satellite formation. Specifically, these variables are number of satellites, altitude and inclination of the

chief orbit, the relative differences between the Keplerian elements of different satellites and strategies for payload pointing for obtaining multi-angular images simultaneously. Three potential strategies or imaging modes are

1. Fixed reference satellite, wherein one satellite always points nadir while others point at the ground spot below the reference satellite

2. Variable reference satellite, which is the same as 1 except that the reference satellite varies

3. Tracking mode, where all satellites track pre-defined ground points as they emerge from and disappear over the horizon.

The third imaging mode obviously provides the most angular coverage, at the cost of spatial coverage because only a small set of ground points can be tracked with one formation of satellites.

We have not optimized the design variables in this paper, instead, have used formation architecture designs that have been shown to be optimum for land surface (not TOA) $BRDF$ estimation from space, as averaged over one day of simulation

(Nag et al. (2015)). The architectures were compared to each other on the basis of root mean squared (RMS) $BRDF$ errors, which were computed against airborne data collected over years of campaigns by the NASA Goddard Space Flight Center's Cloud Absorption Radiometer, or CAR (Gatebe and King (2016)). Since the CAR can fly around any ground spot in circles with adjustable altitude, it can collect hundreds of thousands of angular measurements and estimate $BRDF$ more accurately than any other aerial or space instrument. CAR data can therefore be assumed as a standard to compare other measurement

techniques for $BRDF$ and its dependent products. The airborne data was organized by surface type, whose global distribution was obtained from the MODIS Global Land Cover Facility (Friedl et al. (2010)). The errors were derived from the $BRDF$ retrieved at every instant by the formation, depending on the surface type expected to be seen by the formation at that particular instant.

## 2.2   Information content analysis

We use an Information Content (IC) assessment method that applies Bayesian statistical techniques to connect measurements to the expected retrieval success of geophysically relevant parameters. This technique is described for atmospheric remote sensing by Rodgers (2000), and more specifically for multi-angle polarimetric remote sensing of aerosols by, for example, Hasekamp and Landgraf (2007), Knobelspiesse et al. (2012) and Xu and Wang (2015). This analysis uses the same software for Radiative Transfer (RT) and other computations as Knobelspiesse et al. (2012).

Fig. 4 is a conceptual illustration of the IC analysis technique. We consider two multidimensional spaces. The state (or parameter) space spans possible geophysical parameter values (left box in Fig. 4), while the observation space spans possible


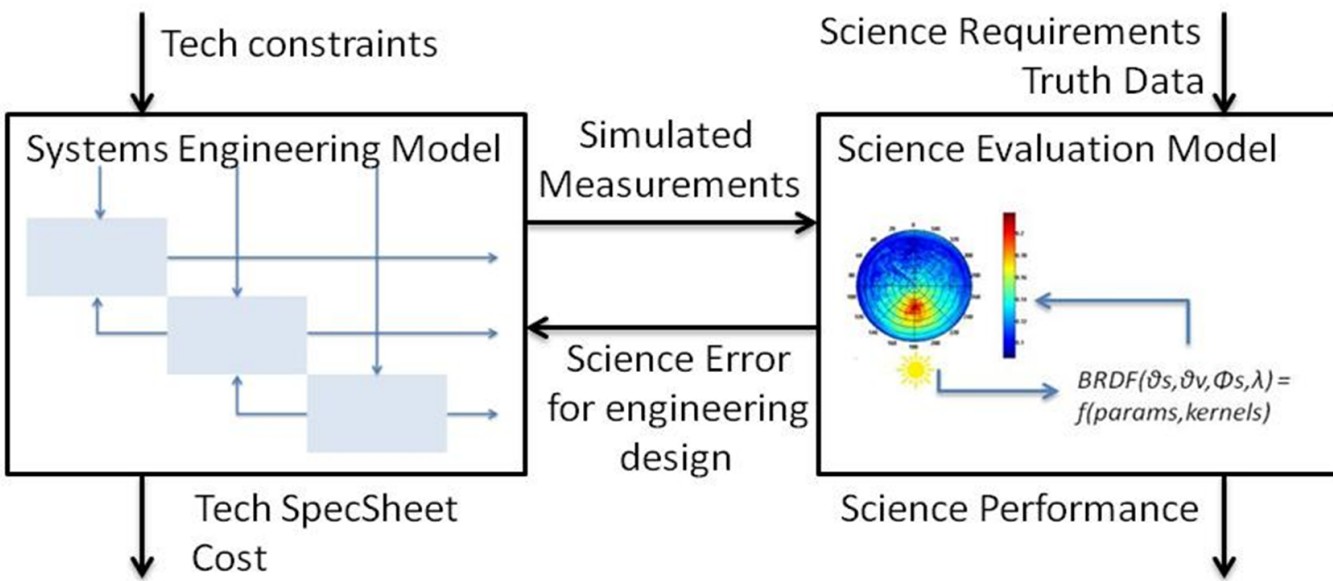

**Figure 3.** From Nag et al. (2016b), the relationship between the systems engineering model and the science evaluation model. The engineering model takes in the mission's technical constraints and outputs optimized technical specifications, simulated measurements and cost for hundreds of simulated architectures. The science evaluation model performs information content analysis on the simulated measurements based on science requirements and reference data, and outputs science performance and error values for each architecture.

observed measurement values (right box). Geophysical reality may be expressed by a point within state space represented by the vector $x$, where each element contains parameters describing aerosol size distribution, refractive index, etc. (see table 2 for a list of parameters used in this analysis). This corresponds to a point represented by the vector $y$ in observation space, where each element contains the measured reflectance or radiance for a particular geometry and spectral channel. Connecting the two is the forward (RT) model, ($F(x) = y$), which produces a simulated observation, $y$, given geophysical parameters, $x$. All measurements have uncertainty, so an observation is really an expression of a volume within observation space, represented by both $y$ and uncertainties about those points $[\Delta y_1, \Delta y_2, ...]$. We are concerned with how that volume in observation space maps to a volume in parameter space, as it shows the utility of a measurement system. This relies on both instrument characteristics and the relationship between state and observation spaces, which we explore with our RT model. We express this by calculating the retrieval error covariance matrix, $\hat{\mathbf{S}}$, :

$$\hat{\mathbf{S}}^{-1} = \mathbf{K}^T \mathbf{S}_\epsilon^{-1} \mathbf{K} + \mathbf{S}_a^{-1}, \tag{2}$$

where $\hat{\mathbf{S}}$ is the uncertainty volume surrounding $x$. The diagonals of this square matrix correspond to squared uncertainties associated with each parameter in $x$, while off diagonal elements are the covariances between them. The retrieval error covariance matrix depends on the observation error covariance matrix, $\mathbf{S}_\epsilon$, the *a priori* error covariance matrix, $\mathbf{S}_a$, and the Jacobian





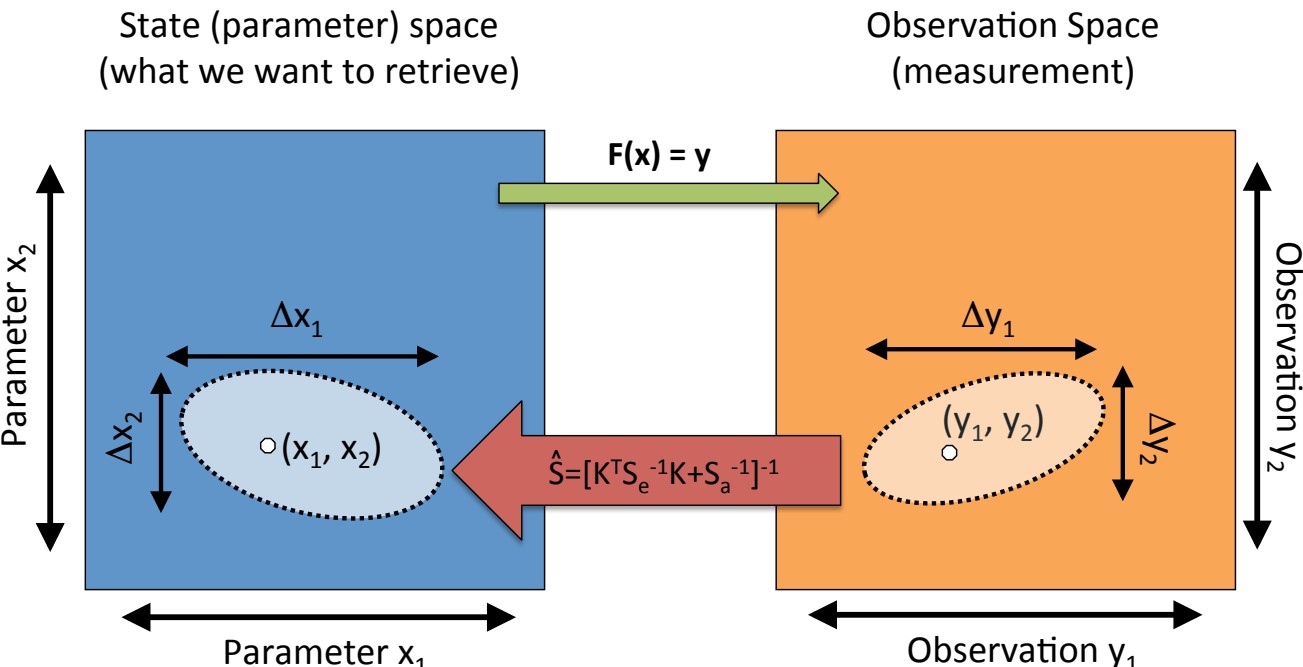

**Figure 4.** This figure is an illustration of Information Content (IC) assessment concepts. We consider a state space (blue) representing all possible geophysical parameter values. Each point within this space is a plausible geophysical state, expressed as the vector $x$. There is a corresponding space representing measurements by an observing system (orange). The vector $y$ in this space represents a measurement. Practically, all measurements have uncertainty, represented by the light orange portion of observation space. We are interested to learn what uncertainty volume (light blue) corresponds to the measurement uncertainty volume in state space. To find this, we use the forward model ($F(x) = y$, green arrow) which produces a simulated observation given a geophysical state. Since our forward model is highly nonlinear, it is not easily inverted to go in the other direction, from observation space to state space. However, the sensitivity of this forward model (the Jacobian, $\mathbf{K}$) if assumed linear for small perturbations, can be combined with knowledge of the measurement uncertainty, $\mathbf{S}_\epsilon$, and prior knowledge of the state space, $\mathbf{S}_a$, to determine the retrieval error covariance matrix, $\hat{\mathbf{S}}$. This is indicated by the red arrow, and must be computed for several locations within the spaces because of the nonlinearity in the forward model to achieve a realistic understanding of these relationships. In practice, we work with systems with much higher dimensionality than two, which is used here for clarity.





matrix, $\mathbf{K}$ ($^T$ denotes the transpose, and $^{-1}$ the inverse). The observation error covariance matrix, $\mathbf{S}_\epsilon$, corresponds to the area surrounding $y$ in Fig. 4, where diagonals are the squared uncertainties of each measurement in $y$, and off diagonal elements their covariance. This matrix contains instrument calibration accuracies, typically the largest source of uncertainty for aerosol remote sensing instruments. The *a priori* error covariance matrix, $\mathbf{S}_a$, represents knowledge of the parameters before a measurement. This is the boundaries of possible state space, the total area (blue in Fig. 4) of that space. It is defined in a similar fashion as $\hat{\mathbf{S}}$ and $\mathbf{S}_\epsilon$, where diagonals are the squared prior parameter uncertainty, and off diagonals their covariance. The Jacobian matrix $\mathbf{K}$ is the forward model sensitivity, estimated with a forward difference:

$$K_{ij}(\mathbf{x}) = \frac{\partial F_i(\mathbf{x})}{\partial x_j} \approx \frac{F_i(\mathbf{x}') - F_i(\mathbf{x})}{x'_j - x_j}, \tag{3}$$

which assumes $F(\mathbf{x})$ is linear for the perturbation $x'$. This is reasonable for our RT model since we use small perturbations, although $F(x)$ is highly nonlinear overall. To compute the Jacobian, we must execute the RT model for $x$, and then again with a perturbation for each element in $x$. Wang et al. (2014) is an example of an information content assessment system that uses a linearization of the forward model to compute the Jacobian, which will be more robust if $F(\mathbf{x})$ is nonlinear over small perturbations. To assess the overall information content of a system we also must compute $\hat{\mathbf{S}}$ using an assemblage of different Jacobians, each partial derivative estimated at various locations within state space. In other words, we compute equation 2 for many different scenes, with different aerosol optical properties, and draw conclusions based on the aggregate result. To some extent, multiple assessments may also cancel inaccuracies due to the linearity assumption over small perturbations as well.

$\hat{\mathbf{S}}$ can also be used to predict the uncertainty of parameters that are not explicitly retrieved, but derived from retrieved parameters (Hasekamp and Landgraf (2007)). If the definition of a parameter, $a$, is generalized such that $a = G(\mathbf{x})$, then the uncertainty for $a$ is

$$\sigma_a = \sqrt{\sum_{i=1}^{n} \sum_{j=1}^{n} \hat{\mathbf{S}}_{i,j} \frac{\partial a}{\partial x_i} \frac{\partial a}{\partial x_j}}. \tag{4}$$

This presumes that $G(\mathbf{x})$ can be differentiated, which in our example is the case (see Section 3.3).

A useful reformulation of $\hat{\mathbf{S}}$ is the averaging kernel matrix, $\mathbf{A}$, which indicates ability to retrieve $x$ given $\mathbf{K}$, $\mathbf{S}_\epsilon$, and $\mathbf{S}_a$. The averaging kernel matrix is

$$\mathbf{A} = \left[ \mathbf{K}^T \mathbf{S}_\epsilon^{-1} \mathbf{K} + \mathbf{S}_a^{-1} \right]^{-1} \mathbf{K}^T \mathbf{S}_\epsilon^{-1} \mathbf{K} \tag{5}$$

$\mathbf{A}$ has the same dimensionality of $\hat{\mathbf{S}}$, with each row and column corresponding to a parameter in $x$. $\mathbf{A}$ is also known as the model resolution matrix, the state resolution matrix or the resolving kernel, although we will use the averaging kernel matrix as the preferred term. A perfect retrieval is indicated if $\mathbf{A}$ is an identity matrix, otherwise diagonal values smaller than one indicate less information about the associated parameter. In other words, it can be approximately considered the fraction of the





result that comes from the observation, not $\mathbf{S}_a$. A useful scalar value determined with the averaging kernel matrix is the degrees of freedom for signal ($DFS$), computed as the trace of $\mathbf{A}$. Since it is a scalar, $DFS$ is a convenient distillation of the ability of a measurement system to retrieve geophysical parameters of a specific scene.

$\hat{\mathbf{S}}$ has useful information in the off diagonal terms on the matrix related to the correlation in the retrieved uncertainties. This can be expressed with retrieval error correlation matrix, $\hat{\mathbf{R}}$, which is computed from $\hat{\mathbf{S}}$

$$\hat{R}_{i,j} = \frac{\hat{\mathbf{S}}_{i,j}}{\sqrt{\hat{\mathbf{S}}_{i,i}}\sqrt{\hat{\mathbf{S}}_{j,j}}}. \tag{6}$$

Correlation strength (values close to 1 or -1) indicate a reduction in the uncertainty volume in State space (i.e. narrowing of the ellipse of the light blue shaded area in Fig. 4), and thus a relationship between parameters that indicates increase in IC compared to uncorrelated parameters.

The information content assessment tools we have described here, while powerful, have a number of limitations and caveats that must be mentioned. We can predict uncertainty for a retrieval, but this assumes we have:

– perfect knowledge of observation uncertainty (and the assumption that such uncertainty is Gaussian),

– perfect forward model simulation of geophysical reality (although Rodgers (2000) does describe techniques to incorporate forward model error if it is known and quantifiable), and

– perfect algorithm ability to converge to the best retrieval from the observations.

Of course, we are far from perfect. This IC assessment technique therefore presents the best case scenario for a given measurement. It is useful because we have a quantitative means to connect the observation and scene conditions to retrieval ability with limited computational expense. This means our assessment is ideal for relative comparisons (minimizing the impact of assumptions) for specific hypothesis tests. As we will describe in more detail later, we test 16 different observation configurations, each with more than a hundred orbital geometries, for six different scenes, for a total of nearly 10,000 individual assessments. We do this to provide a thorough test of the IC contained in small satellites in formation and multi-angle observations on one platform.

We should also note that swath width, spatial resolution and other other details associated with the ability of an observing system to properly sample the global state are not assessed in this analysis. This study can be considered one step simpler than a full blown Observing System Simulation Experiment (OSSE), where a global model of aerosol properties is sampled by an observing system to determine its capability (for example, Colarco et al. (2010)). Our assessment describes the information contained within a single pixel. In this sense, we undertook this work to help decide if the computational expense and methodological complexity of such a study is worth the effort.





## 3  Method

Our hypothesis is that the IC content contained in observations by small satellites in formation flight is comparable to that of multi-angle observations on one platform, where the primary difference is that such observations have a variety of view

zenith and azimuth angles, and are not restricted to one azimuth plane as is the case with a single multi-angle instrument. To test this, we simulate a variety of different observation geometries while keeping all other instrument characteristics (such as spectral sensitivity and measurement uncertainty) the same. Instruments systems with sensitivity to linear polarization are tested along with those that have sensitivity to radiance or reflectance alone (see subsection 3.1 for more details). Using a systems engineering model, we then determine the orbital geometries of each configuration over the course of a day, providing more

than one hundred daytime observation geometries for each configuration (subsection 3.2). Next, we perform RT calculations for each of these cases for six different types of scenes (three over land, three over the ocean, subsection 3.3), then assess the results with IC analysis techniques (subsection 3.4).

### 3.1  Simulated Instrument Characteristics

We simulate between three and nine small satellites in formation flight to compare to a multi-angle instrument with nine view

angles in the along track direction. The small satellites are considered to have a single viewing angle each, while the nine view angles of the multi-angle instrument were chosen to mimic MISR. The MISR instrument observes in the along track direction at $70.5°$, $60°$, $45.6°$, $26.1°$ fore and aft of nadir (a total of nine view angles, including nadir, Kahn et al. (2001)). All instruments are simulated to have the same spectral and uncertainty characteristics. We've chosen to use four narrow spectral channels, centered at 410, 555, 865 and 2,250nm. While no instrument has exactly these channels, many are shared with orbital

instruments such as MISR and POLDER, and prototype designs such as the Multi-Viewing Multi-Channel Multi-Polarization Imager (3MI) or the Aerosol Polarimetry Sensor (APS) (Kahn et al. (2001), Fougnie et al. (2007), Peralta et al. (2007), Marbach et al. (2013)). Two versions of the instrument are assessed. 'Polarimetric' instruments are sensitive to linear polarization in all channels, meaning the first three elements ($I$, $Q$ and $U$) of the Stokes polarization vector (see section 3.3 for radiometric unit definition). Radiometric uncertainty (for $I$) is taken to be 0.03, while polarimetric uncertainty (the uncertainty of the Degree

of Linear Polarization, $DoLP$, the ratio of linearly polarized to total radiation) is 0.005. 'Radiometric' instruments are not sensitive to linear polarization, but $I$ of the Stokes vector alone, for which a 0.03 uncertainty is also assumed. In all cases, uncertainties are Gaussian and completely uncorrelated, such that the off-diagonal elements of $\mathbf{S}_\epsilon$ are zero.

### 3.2  Orbit Design and Systems Engineering

We use the systems engineering model to simulate angular measurements over one day ($>$15 orbits per satellite). For formation

flights by multiple satellites, one satellite in the formation is simulated to point at nadir, while the other satellites point to the ground spot below the first satellite. Payload pointing strategy 2 in section 2.1 is used, i.e. the nadir pointing satellite changed dynamically based on an algorithm documented in Nag et al. (2016b), because this imaging mode was found to produce the least surface $BRDF$ estimation errors without compromising spatial or global coverage. It is assumed that algorithms are



run and decisions made in ground stations and communicated to the satellites during daily overpasses. For a given altitude-inclination combination, previous studies (Nag et al. (2015)) have compared a total of 1,254 architectures containing three to nine satellites in terms of surface $BRDF$ error, averaged (root mean square) over the simulation day. The only orbital difference among the satellites are in their right ascension of ascending node and mean anomaly, because these were found

to be maintainable over a year with propellant available within small satellite of commercial capability (Nag et al. (2016b)). Dependence on altitude and inclination of the orbit was found to be negligible because the planar and in-plane separation of the satellites can be changed in order to achieve similar maximum spreads across orbits. Performance was found to depend largely on the number of satellites and how they are arranged.

**Table 1.** RAAN/Mean Anomaly in degrees for each satellite in the selected formations with respect to the first satellite, and the number of observations in a day with a solar zenith angle, $\theta_s$, less than $85°$. Note that the nine view multi-angle single satellite has 119 observations with $\theta_s < 85°$, which was slightly larger due to a higher orbit (710km compared to 650km).

| | # $\theta_s < 85°$ | Sat 1 | Sat 2 | Sat 3 | Sat 4 | Sat 5 | Sat 6 | Sat 7 | Sat 8 | Sat 9 |
|---|---|---|---|---|---|---|---|---|---|---|
| 3 sat formation | 107 | 0/0 | 5/-4 | 0/5 | | | | | | |
| 4 sat formation | 106 | 0/0 | -5/-6 | 5/-4 | 0/5 | | | | | |
| 5 sat formation | 106 | 0/0 | -5/-6 | 5/-4 | -5/6 | 5/4 | | | | |
| 6 sat formation | 107 | 0/0 | -5/-6 | 5/-4 | -5/6 | 5/4 | 5/-1 | | | |
| 7 sat formation | 107 | 0/0 | -5/-6 | 5/-4 | -5/6 | 5/4 | 5/-1 | -5/1 | | |
| 8 sat formation | 107 | 0/0 | 0/-5 | -5/-6 | 5/-4 | -5/6 | 5/4 | 5/-1 | -5/1 | |
| 9 sat formation | 106 | 0/0 | 0/-5 | -5/-6 | 5/-4 | 0/5 | -5/6 | 5/4 | 5/-1 | -5/1 |

Architectures corresponding to the lowest average (root mean square, RMS) surface $BRDF$ error over time, when compared

to CAR data, are used as case studies in this paper. All the satellites are in a 650 km circular orbit at a $51.6°$ inclination. The relative right ascensions of the ascending node (RAAN) and mean anomalies with respect to the first satellite for each satellite in the six formations are listed in Table 1. The satellites are arranged in one to three orbital planes not more than $5°$ apart in RAAN, for all formations. They can be initialized either by a propulsive launcher or allowed to achieve their final configurations through one to seven months of drifting, depending on the availability of 220 m/s to 10 m/s of correction fuel. More fuel allows

for faster initialization. As confirmed in Nag et al. (2016b), the monthly $\Delta V$ per satellite to maintain the formation can be as low as 0.5 m/s, and more than 80% overlaps between the ground spots are guaranteed for $0.5°$ of pointing control and 2 km of GPS error. For reference, $\Delta V$ (literally "change in velocity"), is a measure of the impulse that is needed to perform a trajectory maneuver in space or at launch. It is a scalar with units of speed and indicates, along with mass and propellant type, the amount





of fuel required to perform the maneuver. In the context of this paper, $\Delta V$ indicates maneuvers to maintain the satellite orbits against gravitational and atmospheric disturbances.

The orbital elements proposed above are achievable within commercial small satellite technology. The results presented in this paper are not dependent on the size of the satellite, which can be scaled up to fit the instruments and associated calibration mechanisms required to achieve aerosol science, without any loss of generality of the presented information assessment.

The inputs (simulated measurements) from the systems engineering model to the information content analysis model, as seen in Fig. 3, are the angles of measurement for the co-pointed ground spot, per satellite and per time step (one minute) for every formation in Table 1. Note that 'Sat 1' is not the reference satellite in terms of pointing, but chosen randomly for relative orbital element representation in Table 1 only. The nadir pointing satellite changes over the course of the simulation and the effective angles automatically calculated in the simulation.

## 10  3.3   Radiative transfer simulation

We use a nested RT model that first computes the single scattering Lorenz-Mie solution to Maxwell's equations for spheres, then incorporates that with other computations for a plane parallel, multiple scattering scene using the Doubling or Adding method (Hansen and Travis (1974)). The software performing these calculations was created at the NASA Goddard Institute for Space Studies (GISS), and has been validated against the results in de Haan et al. (1987) to be within 1% in radiance

(average absolute deviation 0.03%) and 0.08% in $DoLP$ (average absolute deviation 0.02%). This software has been used for general tests of aerosol remote sensing with polarimeters (Cairns et al. (2003)), incorporated into optimal estimation aerosol, cloud and land surface parameter retrieval algorithms (Knobelspiesse et al. (2011a, b); Ottaviani et al. (2012); van Diedenhoven et al. (2012); van Diedenhoven et al. (2014); Ottaviani et al. (2015), and used for information content analyses such as this one (Knobelspiesse et al. (2012), Ottaviani et al. (2013), Knobelspiesse et al. (2015)).

For a given parameter vector, $x$, the RT model produces values of reflectance, $R_I(\theta_v, \theta_s, \phi, \lambda)$ and Degree of Linear Polarization, $DoLP(\theta_v, \theta_s, \phi, \lambda)$, at the specified viewing geometry $(\theta_v, \theta_s, \phi)$ and wavelength $(\lambda)$. Reflectance is computed by normalizing the observed radiance by solar irradiance, sun earth distance, and the cosine of the solar zenith angle, and is unitless (see Knobelspiesse et al. (2012) for more details). $DoLP$ is also unitless, as it is the ratio of the linearly polarized to total reflectance, computed $DoLP = \sqrt{Q^2 + U^2}/I$ (recall that $Q$ and $U$ are the elements of the Stokes polarization vector that

indicate linear polarization). The RT model produces reflectances for the full $BRDF$, and $BPDF$ so our measurement vector, $y$, is simply created with the subset of the $BRDF$ relevant to the geometry of the satellite configuration in question.

We considered two types of scenes, and simulated each with three different levels of aerosol loading. For most cases of multi-angle aerosol property retrieval, the information content contained in a scene depends on instrument configuration, decisions about which parameters to retrieve, and aerosol load, and only weakly on aerosol optical properties (Knobelspiesse

et al. (2012)). A large number of simulations are therefore not required for this assessment. Aerosol loading, described by the Aerosol Optical Thickness ($AOT$) at 555nm, was selected to be $AOT(555nm) = 0.05, 0.15,$ and $0.25$. The lowest $AOT$ value can be considered a low loading at the threshold of detectability, the medium value roughly represents a global mean Remer et al. (2006), while the highest $AOT$ could be considered a moderate to high aerosol load (note $AOT$ is typically log-normally





distributed, O'Neill et al. (2000)). As we shall see in the next section, aerosol retrieval ability increases with aerosol load, so higher values than these would have better retrievals, rare that they may be globally.

**Table 2.** Characteristics of the two scene types selected for simulation with our RT software. Dubovik et al. (2002) was the source of the aerosol optical properties. "Maritime" aerosols represent a mean of optical properties observed by a Cimel sun photometer in Lanai, Hawaii, USA. Surface values for the ocean were chosen so that the $Chl-a$ concentration and wind speed parameters generally represent oligotrophic open ocean conditions. "Continental" aerosols are the mean of observations from the same type of instrument in Greenbelt, Maryland, USA (suburban Washington, DC). Surface $BRDF$ values for land are from an analysis of airborne scanner observations at low altitude near the US Department of Energy's Southern Great Plains site in Lamont, Oklahoma, USA. These observations were made of recently plowed fields, and use the 'sparse vegetation' $BRDF$ parameterization model (Knobelspiesse et al. (2008)). Free parameters are highlighted in bold, and have associated *a priori* values. *A priori* values are the diagonals of $\mathbf{S}_a$, are the one sigma variabilities in the source data used to specify the associated scene parameter. Note that $AOT$, which changes for each test case, is considered a free parameter with an *a priori* value of 0.154. The reference wavelength for $AOT$ is 555nm.

| | Maritime - ocean | *a priori* | Continental - land | *a priori* |
|---|---|---|---|---|
| Fine size mode AOT | **0.018, 0.054 0.090** | 0.154 | **0.045, 0.135, 0.225** | 0.154 |
| Fine AOT fraction | 36% | | 90% | |
| Fine mode Refractive index | 1.37-i0.001 | | 1.40-**i0.003** | i0.01 |
| Fine eff. radius | **0.135**$\mu m$ | 0.035 | **0.170**$\mu m$ | 0.035 |
| Fine eff. variance | 0.193 | | 0.155 | |
| Coarse size mode AOT | **0.032, 0.096, 0.16** | 0.154 | **0.005, 0.015, 0.025** | 0.154 |
| Coarse mode Refractive index | 1.37-i0.001 | | 1.40-i0.003 | i0.01 |
| Coarse eff. radius | **3.36** $\mu m$ | 1.310 | 5.52 $\mu m$ | |
| Coarse eff. variance | 0.704 | | 0.755 | |
| Surface | Chl-a=**0.03**$mg/m^3$ | 5 | parameterization: | |
| | Wind speed=**8**$m/s$ | 2.5 | **2 spectrally invariant** | 0.5, 0.2 |
| | | | **+ 1 for each channel** | 0.05 |
| Total free parameters | **6** | | **10** | |

Table 2 contains details about about each scene type. Both consist of a bimodal aerosol size distribution, partitioned into fine and coarse size modes with identical, spectrally invariant, complex refractive index (as is the case for the *Maritime: Lanai,*





*Hawaii* and *Continental: Washington, DC* aerosol models in Dubovik et al. (2002)). Partitioning between size modes is done in terms of the $AOT(555nm)$ fraction of the fine mode. Thus, the Maritime scene is dominated by aerosols in the coarse size mode, while the Continental scene is dominated by aerosols in the fine size mode, and the total aerosol load ($AOT$) for the scene is simply the sum of the fine and coarse mode loads.

The ocean surface reflectance was parameterized to represent a moderate Chlorophyll-a load (a proxy for phytoplankton

concentration that drives ocean reflectance), and specular reflectance of a surface roughened by a wind speed of 8 m/s, after the model in Chowdhary et al. (2012). The land surface was parameterized using observations from a low altitude aircraft scanner described in Knobelspiesse et al. (2008) as a model. This used the "Ross-Li" surface $BRDF$ parameterization method (Lucht et al. (2000)) for measurements of recently plowed agricultural fields near the US Department of Energy's Southern Great Plains site in Lamont, Oklahoma, USA. Both scene types used a single parameter to represent the $BPDF$ with an angular

dependence similar to Fresnel reflectance (for an example see Waquet et al. (2009)).

The RT model was used to compute the simulated measurement vector $y$ and its perturbation sensitivity as expressed in the Jacobian matrix, **K**. Perturbations (free parameters in a retrieval) were performed for six parameters for the Maritime scene, and ten for the Continental scene. The difference is due to the larger number of parameters needed to describe the land surface scene, including a spectrally variable parameter describing isotropic surface reflectance. Essentially, an additional parameter is needed

for each spectral channel over land. Four parameters are used to describe aerosols in both types of scenes. For the Maritime scene, free parameters are the fine size mode $AOT$, fine size mode effective radius, coarse size mode $AOT$, and coarse size mode effective radius. For the Continental scene (dominated by fine size mode aerosols), free parameters are the fine size mode $AOT$, fine size mode effective radius, fine size mode imaginary refractive index and coarse size mode $AOT$. In both cases, this is fewer parameters than the full set needed to describe the scene, and an acknowledgement of the underdetermined nature

of a retrieval with these instrument configurations. In practice, a retrieval would involve the use of aerosol models or some other means of connecting parameters *a priori* to constrain the search in parameter space. For information content assessment, it is important to select a parameter space of realistically retrievable conditions, which is why we have limited the number of perturbations. Since we compare the information contained in different designs in a relative sense, we are less sensitive to the details of our choices of parameter space as long as they are broadly feasible for all our designs.

Fig. 5 is a sample of the RT software output, where the $BRDF$ of the Maritime-ocean scene with a moderate aerosol load ($AOT(555nm) = 0.15$) is shown in the top row for 410nm (left) and 865nm (right). The $BPDF$ (in $DoLP$) is in the lower row. Note the differences between each plot could contain information about the scene, as well as structure contained within each $BRDF$. Large amounts of structure (such as in the $BPDF$ at 865nm) could mean that there is sensitivity to the distribution of observations throughout the $BPDF$. To know for sure, we must place these results in the context of information content assessment.





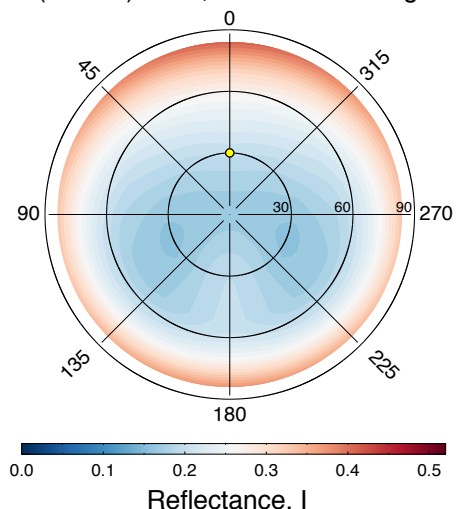

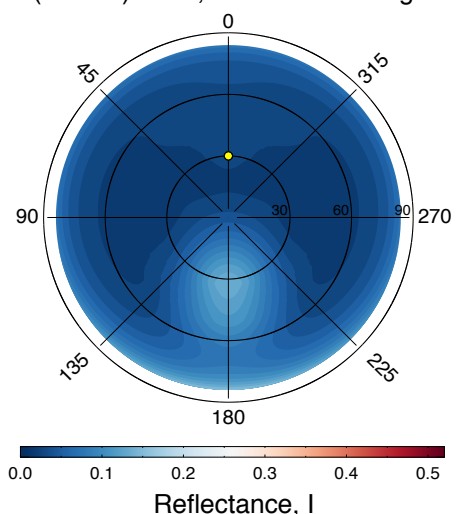

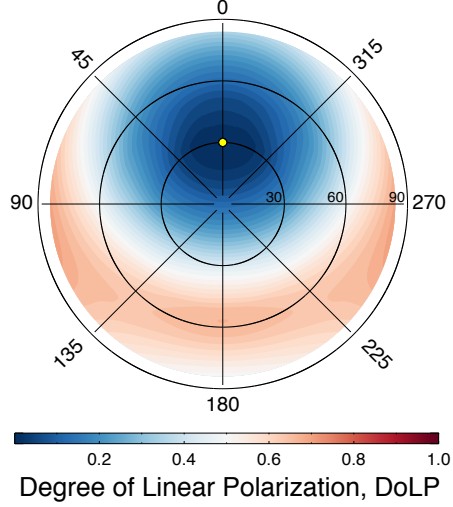

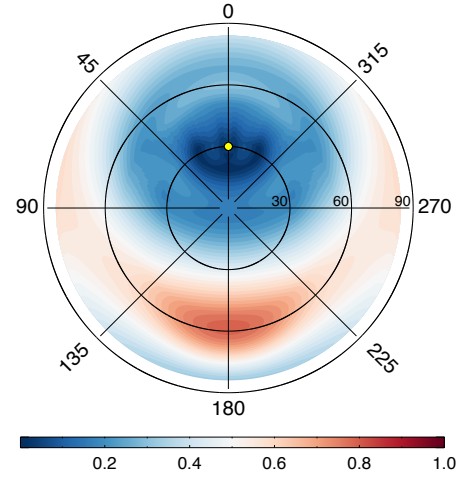

**Figure 5.** Sample Radiative Transfer (RT) software output, for the Maritime-ocean scene described in Table 2 for a moderate aerosol load of $AOT(555nm) = 0.15$. The top row is the $BRDF$ for the ocean and atmosphere scene at 410nm (left) and 865nm (right), while the bottom row is the same for the $BPDF$ (expressed at $DoLP$). Like Fig. 2, view zenith angle ($\theta_v$) is indicated in the radial coordinate dimension, while the relative solar - view angle azimuth ($\phi = \phi_s - \phi_v$) is in the angular dimension, where a value of $0°$ is aligned with the solar azimuth angle. Note the significant differences between each $BRDF$ or $BPDF$, which indicates the structure necessary for parameter retrieval, and the potential importance of appropriate sampling of the $BRDF$ and or $BPDF$ to maximize the information in such a retrieval.




### 3.4 Information content assessment

After completing the steps described above, information content assessment is performed by calculating the retrieval error covariance matrix, $\hat{\mathbf{S}}$ and $DFS$ using the scene Jacobian matrix $\mathbf{K}$ that has been subset appropriately for the instrument design.

We must also create the observation error covariance matrix, $\mathbf{S}_\epsilon$, and the *a priori* error covariance matrix, $\mathbf{S}_a$.

As stated above, $\mathbf{S}_\epsilon$ describes measurement uncertainty, where each diagonal element of the matrix is the square of the individual uncertainty of an observation at the corresponding wavelength, view angle and polarization component. This includes both random and systematic (such as those related to calibration) uncertainties. Off diagonal elements of the matrix represent the correlation between pairs of measurements, which we assumed for these cases is zero, meaning there are no measurement

errors that would simultaneously impact multiple detectors. We expect this to be the case for both observations made by small satellites in formation and by instruments such as MISR, which have independent cameras for each viewing angle. Thus, $\mathbf{S}_\epsilon$ was chosen to be a diagonal matrix, with elements corresponding to $I$ having an uncertainty of $3\%$, and those corresponding to $DoLP$ and uncertainty of $0.5\%$. These values correspond to reasonable radiometric uncertainties for characterized orbital instruments (such as Eplee et al. (2012)), and to desired polarimetric uncertainties cited for most future polarimetric instrument

designs (Kokhanovsky et al. (2015)).

$\mathbf{S}_a$ expresses our knowledge of state space prior to making an observation. In the context of the illustration in Fig. 4, this is the range of state (parameter) space in which a reasonable retrieval solution could reside based on our prior knowledge of the system. Our $\mathbf{S}_a$ was filled with squared *a priori* values shown in table 2, which are based upon the same Dubovik et al. (2002) dataset as the aerosol models themselves. Like, $\mathbf{S}_\epsilon$, we assume no *a priori* correlation between parameters, so $\mathbf{S}_a$ is diagonal.

## 4 Results

Our IC assessment involves the calculation of many (more than a hundred for each scene and instrument configuration) retrieval error covariance matrices, $\hat{\mathbf{S}}$, and the corresponding averaging kernel matrices, $\mathbf{A}$, correlation matricies, and degrees of freedom for signal, $DFS$. We consider eight architectures (the nine view multi-angle instrument plus formations of three through nine single view instruments), for six types of scenes (One maritime, one continental, at three $AOT$ values each), for both

radiometric and polarimetric sensors. In the case of the Maritime-ocean scene, $\hat{\mathbf{S}}$ is a 6x6 matrix, while for the Continental-land scene it is 10x10, meaning an iterative retrieval for those cases would have 6 and 10 free parameters, respectively.

Because of the scale of our IC assessment results, we present a subset that illustrate the overall outcome in light of our goal to compare observations by formations of single view instruments to a multi-angle instrument. We start by comparing the degrees of freedom for signal ($DFS$, Section 4.1) for different instruments and scenes as an overall metric of IC. Next, we

assess the uncertainty for two aerosol parameters: the $AOT$ (Section 4.2) and the fine size mode effective radius (Section 4.3). These were chosen because they were free parameters in all simulated scene types, and because they are common to many aerosol retrievals. Section 4.4 describes the results in terms of the diagonals of the averaging kernel matrices, while in (Section 4.5) we investigate the retrieved parameter correlation, which is not expressed in either the $DFS$ or the individual parameter uncertainties.



## 4.1 Degrees of Freedom for Signal

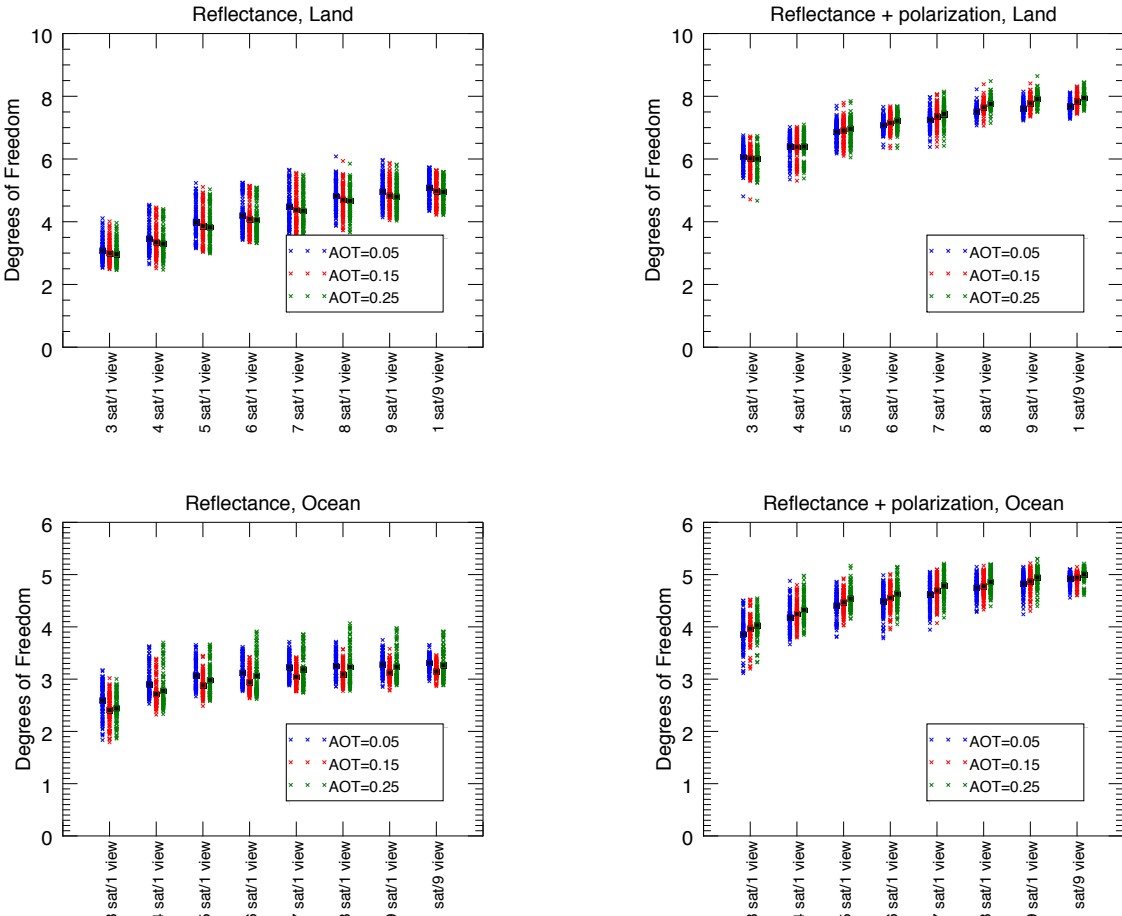

**Figure 6.** The degrees of freedom for signal ($DFS$, described in Section 2.2) is plotted for continental aerosols over land (top row), maritime aerosols over ocean (bottom row) for instruments using reflectance alone (left column) and reflectance plus $DoLP$ (right column). The simulated aerosol load is indicated by color and position, where total $AOT(555nm)$ equal to 0.05 is blue (left), $AOT(555nm)$ equal to 0.15 is red (center), $AOT(555nm)$ equal to 0.25 is green (right). The number of single view satellites is indicated along the abscissa, with the exception of the nine angle "multi-angle" instrument at the far right of each plot. The ordinate axis is the $DFS$, with a range representative of the theoretical maximum for that retrieval. The maximum $DFS$ is equal to the number of free parameters (see Table 2), and is larger over land than over the ocean because of the larger number of parameters required to describe land surface reflectance. Thus, $DFS$ indicates ability to retrieve both aerosol and surface parameters simultaneously.





As described in Section 2.2, the $DFS$ is the trace of the averaging kernel matrix and therefore represents the overall capa-

bility of a measurement system. Capability, in this sense, is combined for parameters of interest to us (descriptive of aerosols) and those required to constrain surface reflectance. Fig. 6 presents the $DFS$ for each simulated scene and instrument configuration. In each plot, instrument configuration is expressed along the abscissa, and the $DFS$ on the ordinate axis. Instruments using reflectance alone are indicated in the two plots at left, while those using reflectance and $DoLP$ are at right. Aerosol retrievals over land are in the top row of figures, those over oceans are in the bottom row. Scene $AOT$ is indicated by color

and relative position within each plot ($AOT = 0.05$ : blue, left, $AOT = 0.15$ : red, center, $AOT = 0.25$ : green, right). The vertical spread of $DFS$ for each configuration and scene represents the impacts of observation geometry variability among the predicted orbits. Black squares indicate the mean value of each group.

Regardless of scene type, all plots show a gentle increase in $DFS$ as the number of satellites in each configuration are increased. $DFS$ for the nine satellite configuration and the multi-angle satellite with nine viewing angles are nearly indis-

tinguishable, with differences in the mean values well within the variability range due to geometric differences in the orbit. This indicates that the capability of a measurement system, at least as expressed by the $DFS$, are primarily governed by the number of viewing angles, but not how those views are distributed within the $BRDF$ or $BPDF$ (although views from both the multi-angle satellite and the small satellites flying in formation are widely distributed throughout the $BRDF$ or $BPDF$). Furthermore, this figure shows that the number of view angles gradually increases the $DFS$, such that a seven or eight satellite

configuration is nearly as capable as the nine satellite configuration or the nine view multi-angle satellite configuration. For scenes over the ocean, in fact, the $DFS$ tends to level off after five or six satellites. This would indicate that only that many view angles are required, at least as expressed by the $DFS$.

As expected, instrument configurations that utilize polarization have greater $DFS$, since they have access to more information. In fact, polarimetric observations over the ocean have a $DFS$ of nearly five, almost the theoretical limit (six) for that

type of retrieval. We also don't see a large influence of the simulated $AOT$ on the $DFS$. Since ability to retrieve aerosol optical properties depends on the aerosol load itself (Knobelspiesse et al. (2012)), this probably means that the uniform DFS is expressing the transition between strong surface parameter capability when the aerosol load is low, which decreases with a corresponding increase in aerosol parameter capability as the aerosol load increases. We will find further support for this below.

**4.2  Aerosol Optical Thickness**

The $AOT$, as a measure of aerosol load, is one of the primary parameters retrieved from an instrument system. Our analysis expects the retrieval algorithm to independently determine the fine and coarse size mode properties, including the individual mode optical thickness. The total $AOT$ is a simple summation, so the uncertainty in its retrieval can be easily computed via Eq. 4. This is shown in Fig. 7 as a relative (to the simulation $AOT$) uncertainty. Like Fig. 6, this four panel figure shows retrievals over land at top, over ocean at bottom, with reflectance only at left, and reflectance with $DoLP$ at right. Instrument configuration is the abscissa, and relative uncertainty for the total $AOT$ is the ordinate axis. Simulations with a total $AOT$ of



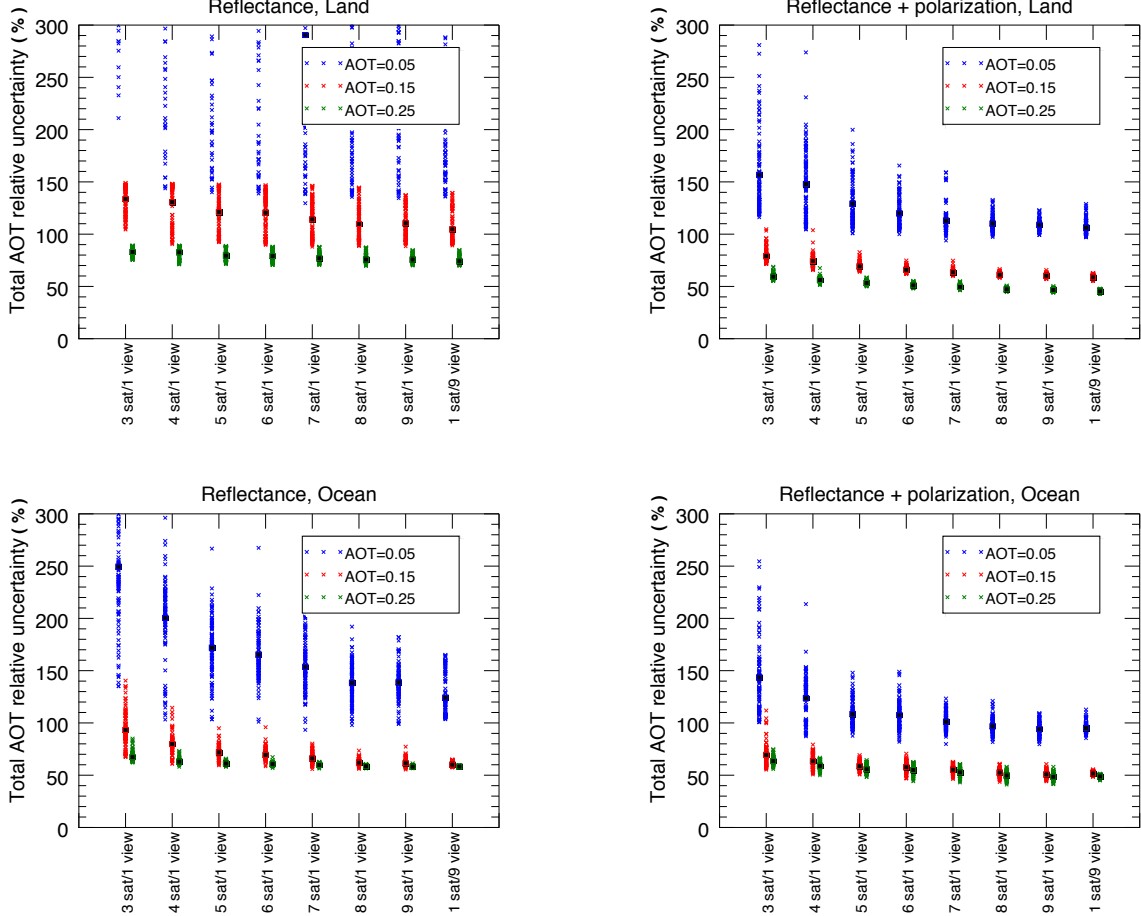

**Figure 7.** Aerosol Optical Thickness ($AOT$) relative uncertainty at 555nm is plotted for continental aerosols over land (top row), maritime aerosols over ocean (bottom row) for instruments using reflectance alone (left column) and reflectance plus $DoLP$ (right column). The simulated aerosol load is indicated by color and position, where $AOT(555nm)$ equal to 0.05 is blue (left), $AOT(555nm)$ equal to 0.15 is red (center), $AOT(555nm)$ equal to 0.25 is green (right). The number of single view satellites is indicated along the abscissa, with the exception of the nine angle "multi-angle" single instrument at the far right. The ordinate axis is the relative uncertainty of the total (fine and coarse size mode) $AOT$.



0.05 are in blue, 0.15 in red, and 0.25 in green. We use 555nm as the reference wavelength for $AOT$. At other wavelengths the $AOT$ may be different, depending on aerosol properties.

Unlike, Fig. 6, however, the $AOT$ relative uncertainty is strongly dependent upon the simulated $AOT$ value itself. This is to be expected, as there is naturally more capability to determine aerosol optical properties if there are more aerosols present to affect the scene. In fact, relative uncertainty for $AOT$ is greater than 100% for nearly all instrument configurations for simulated scenes with an optical depth of 0.05. Considering that the global mean value of $AOT$ is probably three or four times larger (Remer et al. (2008)), this result shows an acceptable lower limit of aerosol detectability. Another striking characteristic

of these results is that the number of viewing angles does not dramatically improve the relative $AOT$ uncertainty, except for the lowest optical depths. Relative uncertainty seems to reach a minima as the number of viewing angles and the simulated $AOT$ increase. An interpretation of this could be that $AOT$ is expressed smoothly and uniformly throughout the $BRDF$, and increasing the number of viewing angles does not add to the information about $AOT$ in the overall measurement. Echoing other analyses, polarization improves the $AOT$ uncertainty, especially over land (Hasekamp (2010); Hasekamp and Landgraf

(2007); Knobelspiesse et al. (2012)).

These results support our hypothesis that single view satellites in formation flight are equally capable as multi-angle observations on an individual satellite, provided that the number of viewing angles are the same. Furthermore, loss of one or more single view satellites does not contribute much to an increase in uncertainty.

### 4.3   Fine size mode effective radius

The uncertainty of determining the effective radius (one of two parameters defining size distribution) of the fine (small) aerosol size mode is plotted in Fig. 8. Like Fig. 6 and 7, this four panel figure shows retrievals over land at top, over ocean at bottom, with reflectance only at left, and reflectance and $DoLP$ at right, while instrument configuration is the abscissa, and relative uncertainty for the total $AOT$ is the ordinate axis. The maximum value of the ordinate axis is the *a priori* uncertainty, which is the theoretical maximum (least certain) value for uncertainty for a parameter in Eq. 2. Results close to this value indicate

that the measurement has provided no additional information about that parameter compared to what was known prior to measurement.

We chose to display the fine mode effective radius because it is a parameter that was shared between both types of scenes, although the fine size mode contributed different amounts to the total $AOT$ in each scene. For ocean scenes, the fine mode contributed 36% to the total $AOT$, while over land the contribution was 90%. This means the fine size mode had a stronger role

modifying the observed $BRDF$ and $BPDF$ over land than over ocean, contributing to the lower uncertainties for the former compared to the latter. Otherwise, the uncertainty for the fine mode effective radius follows many of the same patterns as the $AOT$. For the lowest simulation $AOT$ (0.05), uncertainty was close to the *a priori* value for the reflectance only instruments, but slightly better for instruments that used polarization. Additional angles do help a bit more than was the case for $AOT$, although the improvement is gradual. Furthermore, we found no significant differences between the nine satellites in formation flight compared to a multi-angle satellite with nine views.





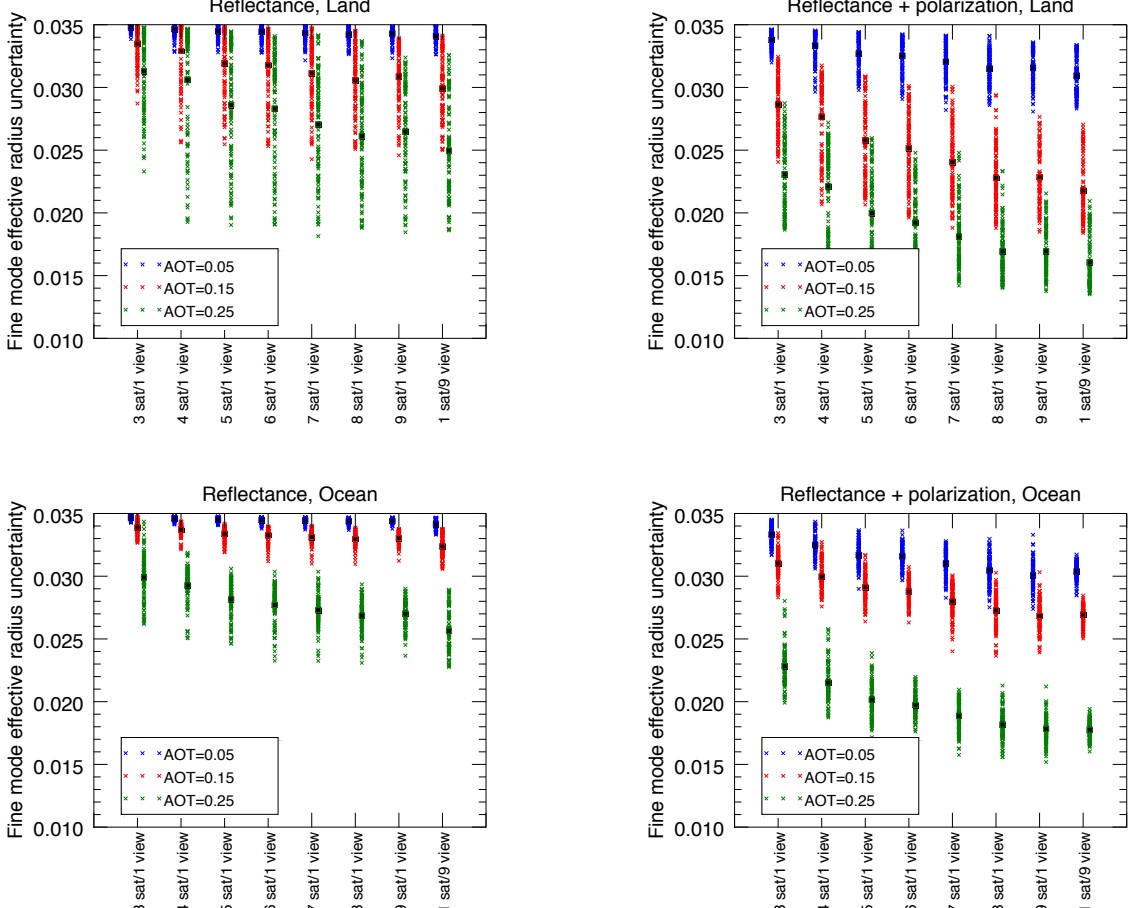

**Figure 8.** Uncertainty in the effective radius for the fine aerosol size mode is plotted for continental aerosols over land (top row), maritime aerosols over ocean (bottom row) for instruments using reflectance alone (left column) and reflectance plus $DoLP$ (right column). The simulated aerosol load is indicated by color and position, where $AOT(555nm)$ equal to 0.05 is blue (left), $AOT(555nm)$ equal to 0.15 is red (center), $AOT(555nm)$ equal to 0.25 is green (right). The number satellites and view angles is indicated along the abscissa. The ordinate axis is the uncertainty for the fine size mode effective radius, from the square root of the corresponding element on the diagonal of $\hat{\mathbf{S}}$. The maximum value of each ordinate axis is the *a priori* uncertainty specified in $\mathbf{S}_a$, the theoretically largest value possible. Results close to this indicate no sensitivity. Also, note that the contribution of the fine size mode to the total aerosol extinction was different for the simulations over land and ocean. Over land, the fine mode contributed 90% to the total aerosol optical depth, while over ocean it was 36% (see Table 2).



### 4.4 Averaging kernel matrix

The averaging kernel matrix (**A**) diagonals for different scene types and observation configurations are illustrated in Fig. 9.

As described in Section 2.2, the diagonals of the averaging kernel matrix represent how independent retrieved parameters will be from the *a priori* matrix. Thus, a diagonal value close to one indicates significant information about that parameter in the observations, or at least that there is significantly more information than defined in the *a priori* matrix. The $DFS$ in Fig. 6 is the sum of these values (in other words, the trace of **A**). This figure thus describes how the $DFS$ are shared among the parameters, an important distinction.

The most obvious inference from Fig. 9 is that values for a parameter are generally equivalent across instrument configurations, with limited improvement as the number of viewing angles increases. Some parameters have high values in nearly all cases, examples include chlorophyll-a for ocean scenes, or the Fresnel polarimetric surface coefficient for land scenes with polarimetrically sensitive instruments. In these cases, *a priori* values were set to be large so that they did not impact results for aerosol related parameters. Retrieval uncertainties for Chlorophyll-a, at least, are not much larger than typical values (McClain

(2009)), reasonable given the low simulated Chlorophyll-a value and lack of ocean color specific channels in our hypothetical instruments. Some parameters have very low values, such as, for land scenes, the imaginary part of the fine size mode refractive index (associated with aerosol absorption) or the Fresnel coefficient for instruments without polarization sensitivity. These parameters show little improvement with additional view angles. Many parameters are in between these extremes, and these show the most sensitivity to an increase in the number of view angles. In any case, this provides the means to understand

the partitioning of degrees of freedom for a given system. For example, a nine view, reflectance only, ocean observation has a $DFS$ of roughly three, which is primarily driven by chlorophyll-a (ocean body reflectance), and the $AOT$ of both aerosol size modes.

These results represent the mean value of **A** diagonals across all orbits for an $AOT(555nm) = 0.15$. For brevity, we have not included results for simulations for smaller and larger $AOT$ since the general patterns remain. As expected, at low $AOT$

values for surface parameters increase while aerosol parameters decrease. It is the opposite for larger $AOT$, as the increase in aerosol load increases the impact of aerosols on observations as the expense of surface reflection.

Finally, what is clear from Fig. 9 is that configuration differences between the 9 satellites flying in formation and the multi-angle single view satellite have an imperceptible impact on **A**.

### 4.5 Correlation matrix

Fig. 10 contains the retrieval error correlation matrix, $\hat{\mathbf{R}}$ (Eq 6), for the nine view angle instrument configurations. As described in Section 2.2, large off diagonal values indicate a smaller volume in retrieval State space, an indication of higher information content for that pair of parameters. We can see this in the slight increase in correlation (or anti-correlation) for retrievals utilizing polarization, an expected improvement with additional knowledge. All scenes demonstrate a strong anti-correlation between $AOT$ and the effective radius of the same size mode. Most importantly, these matrices are nearly identical for the



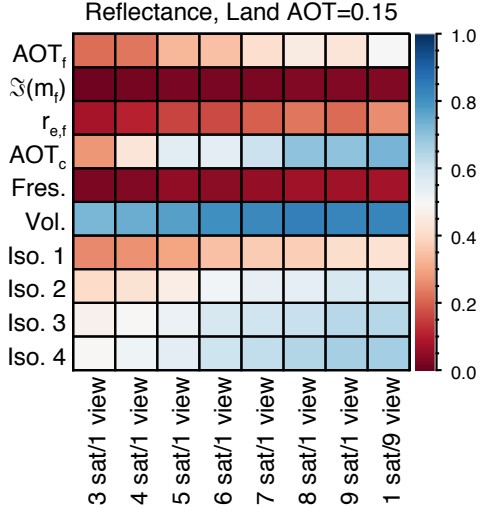

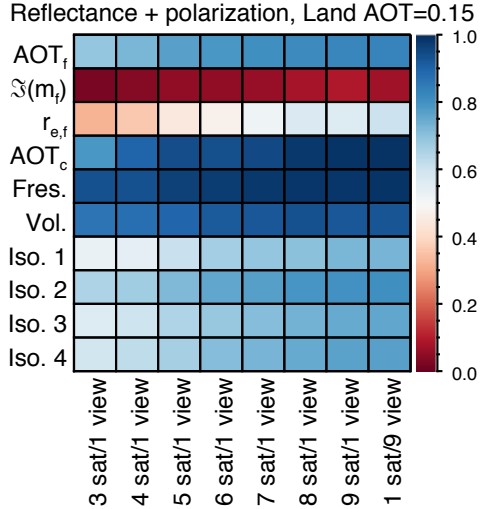

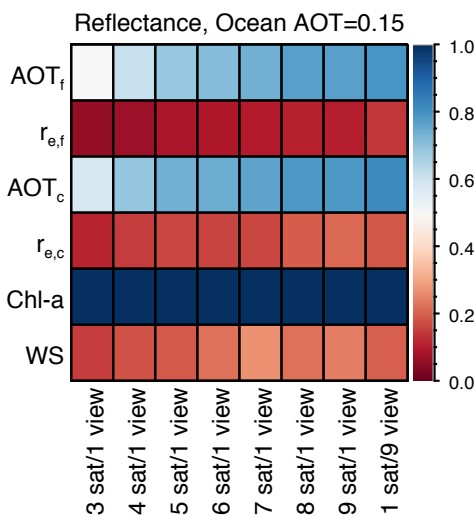

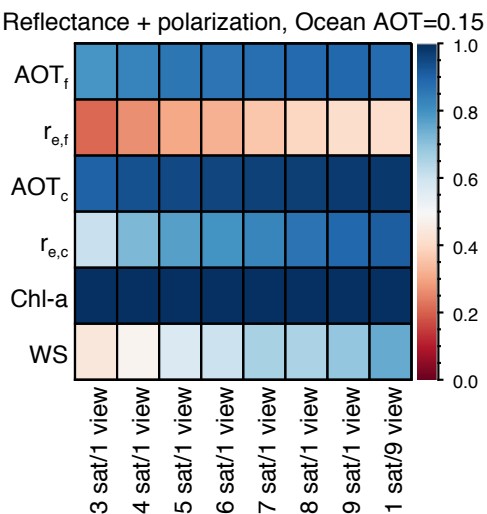

**Figure 9.** Diagonal values for the the mean averaging kernel matrices, **A**, for land scenes (top), ocean scenes (bottom) using observations of reflectance only (left) and $DoLP$ with reflectance (right). These values roughly represent what fraction of a retrieved parameter is due to the observations, and not *a priori* values. Results are shown for each satellite configuration (abscissa), while the ordinate axis has the results for each parameter described in Table 2. For the land scene, these parameters are, from the top: fine size mode $AOT$, imaginary component of the refractive index of the fine size mode, effective radius for the fine size mode, coarse size mode $AOT$, fresnel (polarized) surface reflectance coefficient, volumetric ($BRDF$ shape) surface reflectance coefficient, then isotropic reflectance coefficients for each band, at 410, 555, 865 and 2,250nm. For the ocean scene, these parameters are, from the top: fine size mode $AOT$, effective radius for the fine size mode, coarse size mode $AOT$, effective radius for the coarse size mode, ocean body Chlorophyll-a content, and ocean surface wind speed. The sum of each column is the $DFS$ for each instrument configuration as shown in Fig. 6. This figure therefore displays how those $DFS$ values are partitioned among the retrieval parameters.





**Figure 10.** Correlation matrixes for scenes with the medium simulated optical depth ($AOT(555nm) = 0.15$) are shown for scenes over land (top), over ocean (bottom), for retrievals using reflectance alone (left) and reflectance and $DoLP$ (right). Matrices represent the average for all simulations throughout the orbit for that configuration, although only simulations with nine view angles (nine satellites in formation and nine view multi-angle satellite) are shown.





nine satellites flying in formation flight and the nine view multi-angle instrument. This is further support for the hypothesis that satellites in formation flight are equally capable of retrieving aerosol parameters as multi-angle instruments.

## 5   Conclusions

Our central hypothesis is that aerosol remote sensing is performed equally well by the geometric distribution of observations by small satellites flying in formation and multi-angle views on a single satellite. The main difference between the two types of observations is that multi-angle views on a single satellite are restricted to a single azimuth plane, while small satellites flying in formation observe at a variety of azimuth angles. Such systems therefore sample the BRDF or BPDF in different ways. To test this hypothesis, we have generated a variety of observation formations using a systems engineering orbital model constrained to feasible satellite bus configurations. The geometries of these formations where then used as inputs to an information content analysis, which determines geophysical parameter retrieval capability. This capability was tested for the aggregate of the observation formations for a variety of realistic atmospheric aerosol scenes over land and ocean. These tests were performed for formations of between three and nine satellites to compare to a multi-angle satellite with nine views. All instruments were simulated with identical spectral and measurement uncertainty characteristics. Details about the limitations of our information content technique are discussed in Section 2.2.

The information content analysis reveals that there is no difference between the capability of multi-angle satellite instruments on a single platform compared to an equal number of views from satellites flying in formation. This equivalence is maintained for a variety of aerosol classes, quantities, and scene types (over land or over ocean). The primary factor affecting capability (other than spectral characteristics and measurement uncertainty, which we did not vary) is the number of viewing angles in a observation, and not their distribution throughout the $BRDF$ and $BPDF$. This can be explained by the smooth nature of TOA $BRDF$ and $BPDF$ (see Fig. 5), meaning that constraining observations to a particular plane in the $BRDF$ or $BPDF$ (as is the case with multi-angle instruments) yields no advantages or disadvantages.

We also found that the information content improves only incrementally as the number of viewing angles increases. For some situations and parameters, additional viewing angles provide no improvement after a half dozen or so, while others (typically those for which the observation system has marginal overall information content) do show improvements that eventually level off with many view angles. This is slightly lower than the conclusions of Hasekamp (2010), who found that sixteen viewing angles are sufficient for retrieval of most aerosol optical properties, and that the capability (for aerosols) with more viewing angles is constrained by the angle to angle measurement correlation present in most multi-angle imaging systems. While we do not account for observation correlation (unlikely in measurement systems such as ours) the difference is probably due to our more constrained parameter space.

In addition to our central hypothesis, this analysis reveals useful general details about the information content of multi-angle and multi-angle polarimetric observations. As illustrated in Fig. 6, a multi-angle observation over land has roughly four $DFS$, while polarimetric observations (with a $DoLP$ accuracy of 0.005) add roughly three more $DFS$ to this. Over the ocean, there are roughly three degrees of freedom, and adding polarization about 1.5 $DFS$ to that. Aerosol retrievals require





parameterization of the surface reflectance, so these $DFS$ are partitioned between aerosol relevant parameters and surface relevant parameters, as shown in Fig. 9. The scene aerosol load (total $AOT$) drives this partitioning, such that large $AOT$ increases $DFS$ in aerosol parameters at the expense of surface parameters, and vice versa for small $AOT$. The impact of this can be seen in the individual uncertainty estimates for total $AOT$ (Fig. 7) and fine size mode effective radius (Fig. 8), which show a distinct improvement with increasing $AOT$. These results mirror other sensitivity studies (such as Hasekamp (2010); Hasekamp and Landgraf (2007); Hasekamp et al. (2011); Knobelspiesse et al. (2012)), and supports the notion that our methodology is sound.

To date, multi-angle remote sensing of aerosols have only been performed with instruments that make all of their observations on a single spacecraft. Ongoing technological development means that coordinated observations by formations of satellites are becoming a reality. We have demonstrated that the information contained in such observations would be equivalent to a single multi-angle instrument for aerosol remote sensing. While many technical and scientific matters must still be resolved, this provides an opportunity, as these formations may have engineering, cost or other advantages. They may, for example, be more resilient. Our results indicate that the loss of one or more individual satellites does not dramatically impact the information content in the observation, providing for an opportunity to replace lost satellites, ultimately improving observation continuity. There remains many aspects of such observations to explore, they hold promise for the future of aerosol remote sensing.

*Author contributions.* Author Knobelspiesse performed the information content analysis with input from the systems engineering tool developed by author Nag. Concept, experiment design, and manuscript preparation were joint efforts of both authors.

*Competing interests.* No competing interests are present

*Acknowledgements.* The first author was supported in this research by an award from the NASA New (Early Career) Investigator Program in Earth Science, NNH13ZDA001N-NIP, managed by Ming-Ying Wei and Lin Chambers. The research was conducted at both the NASA Ames Research Center in Moffett Field, California, and the NASA Goddard Space Flight Center in Greenbelt, Maryland. The doubling and adding radiative transfer code used in this work was developed at the NASA Goddard Institute for Space Studies, with recent updates by Brian Cairns and Jacek Chowdhary.




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
