# Peer review of "Remote sensing of aerosols with small satellites in formation flight"

_Atmospheric Measurement Techniques, 2017_

## Referee Comment (RC1) · O. Coddington (Referee) · 4 Mar 2018

Remote sensing of aerosols with small satellites in formation flight
By K. Knobelspiesse and S. Nag
AMTD https://doi.org/10.5194/amt-2017-473

This article assesses the potential for aerosol remote sensing from a formation flight of small satellites via comparisons with a multi-angle single platform (MISR-like) satellite. The authors present their case that a formation of small satellites, each with a single view angle, could perform as well for aerosol remote sensing as a single platform, multi-angle, satellite with the added bonus that small satellites in formation could be replaced as they age at lower overall cost. They use a combined systems engineering approach and information content analysis to support their conclusions.

The authors performed a large amount of simulations for this study: they tested "16 different observation configurations each with more than a hundred orbital geometries, for six different scenes, for a total of nearly 10,000 individual assessments" [line 20, pg 10]. Due to limitations and assumptions in the information content analysis, the authors state their assessment is "ideal for relative comparisons" of information content between a formation of single satellites (each with a single view angle) and a multi-angle satellite. The authors also state that their primary focus is on measurement geometry for determining the portion of parameters that are assumed to be representing aerosol (line 3, pg 4).

In light of the intended impact of the paper, I focus my comments on the orbital geometry findings and the information content assessments. However, it's clear from the text and the results that aerosol remote sensing is very challenging and the retrievals are underdetermined. I admit it's possible that I've missed important and subtle concepts regarding the physics of aerosol remote sensing or misconstrued them.

Comments on the Systems Engineering Aspect

Here, I focus on the pointing control of the formation flight small satellites because I am not finding a clear relation between the orbital geometry metrics presented in the text and how those uncertainties propagated into the information content analysis.  I'm no expert in orbital mechanics, so perhaps I missed it, or did not understand what I read.

*Attitude vs Position*: The authors make the point that the relative positions of the formation flight satellites are less important than their relative attitudes. I believe this makes sense because for aerosol remote sensing, what you are aiming to achieve is a measurement of the spectrum of reflected light at enough view angles – over the same patch of surface area – to observe the BRDF or BPDF function. Thus, the ability of the satellites to be pointed at the same patch of surface area is more important than position control, although the satellites would have to be near enough to each other at a given instance of time to be able to point at the same surface patch, of course.  Conversely, a situation where very tight position control of formation flight satellites would be more important could be where the apertures of individual instrument are combined (in processing analysis) to make a "pseudo" large aperture for viewing very distant objects.

In light of the above, could you explain how the RAAN and Mean Anomaly uncertainties presented in Table 1 that were propagated in the systems engineering model into root mean square (RMS) differences between predicted BRDF and the "truth" BRDF provide a metric of pointing control? My current interpretation is that the RAAN and Mean anomaly uncertainties are rather supporting the case that the small sats can be controlled in close enough formation to be able to point at the same surface patch, not that they actually did point at the same surface patch.

I did note some capabilities listed on page 12 – "adjustments to maintain the formation can guarantee > 80% overlap between ground spots for 0.5 degrees of pointing control and 2 km GPS error."  It's this "agility" aspect, and its associated uncertainty of 0.5 degrees of pointing control, that I'm not finding was propagated into the BRDF/BPDF (p6). Was it and how so (and how did the implementation in the analysis differ for the formation flight satellites as opposed to the multi-angle sensor)? Are the capabilities listed in the paper based on measurement requirements for aerosol remote sensing?

*Observing geometries*: I observe some characteristics in Figure 2 that were not discussed in the paper. Firstly, when I compare the simulated geometries of the formation flight satellites (left plot) with the multi-angle satellite (right plot), I notice that the formation flight satellites did not cover the negative view zenith angles beyond approximately -60 degrees. Does this meet BRDF measurement requirements for aerosol remote sensing (referred to on pg 5), and what are these measurement requirements (list in paper or provide citation)? In the subsequent analysis, how does that absence of observations factor into the RMS differences in observed (simulated) BRDF relative to the "truth" airborne CAR measurements?

Secondly, I also notice that the flight formation architectures have viewing geometries under overhead sun conditions (SZA = 0), whereas the multi-angle satellite configuration did not.  Are there benefits for aerosol remote sensing that you can identify that would be aided by overhead sun angles? I could imagine that there may be benefits and limitations. For example, measured signals would be larger, but perhaps the surface component of that signal becomes even more dominant than the aerosol component?

Finally, because the study's strength is in relative comparisons between formation flight architectures and a multi-angle sensor, it's fair to ask if these differences were incorporated into the subsequent information content assessments, or if the analysis was performed such that the set of viewing geometries and solar zenith angles were first pre-filtered to those common to both small satellites and multi-angle sensor prior to doing the information content assessments.  The text, however, does make clear that spectral sensitivity and measurement uncertainty assumptions have been kept the same for both small satellites and multi-angle sensor measurement error assumptions.

Regarding Figure 3, I don't understand the difference between "science performance" and "science error for engineering design"? Also, it's a small thing, but what is the yellow symbol below the BRDF plot –is it the sun?

Comments on the Information Content Analysis

The authors utilized a statistical technique derived from general inverse theory commonly called optimal estimation [Rodgers, 2000] in their analysis. This approach "connects

measurements to the expected retrieval success of geophysically relevant parameters" (pg 6). Optimal estimation describes the mean and standard deviation of retrieved parameters, based on the information provided in a measurement, apriori information and a model, when one makes the assumption that the forward model relating the parameters and measurements is linear and the measurements themselves are linear (Gaussian) distributed. The authors acknowledge that the forward model for aerosols is highly nonlinear, but that for small perturbations of parameters in the forward model, the sensitivity of the modeled values to that parameter perturbation will be linear (p9). Hence, to build up information content for a broad space of parameter ranges, a number of sensitivity analysis of modeled values to linear parameter perturbations must be created for different locations in the parameter spaces. The "retrieved" parameters are defined by a mapping from a volume in the measurement space to a volume in the parameter space. The forward model, representing aerosol physics as best known, provides the mapping between measurement space 'y' and parameter space 'x'- for the constructed sensitivities (Jacobians) of the measurements to the parameters.

It's too simplistic to represent the forward model as 'F(x) = y' [line 4, pg 7] and in Figure 4. The forward model will also require some assumed model inputs, some (all?) of which you list in Table 2 (the non-bolded values) that will have their own set of uncertainties. Let's call the necessary, but assumed, model inputs 'b'. So, the mapping from observation space, 'y', to parameter space 'x' will also rely on the accuracy of these 'b' values (line 8-9, p7), and how well natural variability in these 'b' values is captured.

I'm missing discussion of the physics of aerosol retrievals and what parameters are retrieved as opposed to what parameters are derived from the retrievals. I initially read Table 2 such that bolded variables were the retrieved parameters. However, discussion of AOT (centered around Equation 4) suggests that AOT is actually a derived parameter.

Then, regarding Equation 4, while I do understand that it has been applied in other studies under presumptions that variables can be differentiated I feel I should point out that it's a weakness in the information content analysis. I'm not trying to lessen your efforts or look for a citation, but I've recently begun grappling with those kinds of challenges and made some advances in quantifying the information that is shared between physical variables (cloud optical thickness and particle size in cloud remote sensing) that wouldn't need to rely on those presumptions and I wanted to point it out to you for your interest. Here I'm speaking of mutual information content metrics that I think would hold promise for aerosol remote sensing challenges (see for example, Section 6.1 in Coddington et al., [2017], J. Geophys. Res. Atmos., 122, 8079–8100).

The assumption that Sa is diagonal (i.e. no a priori correlation between parameters) (Line 19, pg 17) would seem to be weakly supported, given your results in Figure 10. Is this a common assumption in aerosol information content analysis? The same assumption for Se (measurement uncertainty) seems reasonable.

I'm missing a clear definition of how you use the term "scene" (ex. Line 14, pg 9). Do you mean different surface types, or do you mean a variety of different atmospheric conditions for the same surface? If the latter, would "state space" be a better word choice than "scene"? Based on your decision, you might need to do a word search through the paper to find instances of "scene". Then a follow up question, regarding the iterative computation of

Equation 2 for the different scenes: is the a priori error covariance matrix (Sa) held fixed for all scenes? Based on Table 2, I believe that is the case, however, the Jacobian (K) would change.

It's probably a small thing, but I'm actually not sure what is meant by "perfect algorithm ability to converge to the best retrieval from the observations" (Line 16, pg 10). Is it just that you are referring to a retrieval with multiple local minima and you're looking for the "best" one (where "best" is ideally the correct solution).

Lines 23-29, pg 15: You are performing an information content assessment without using an aerosol model to constrain the parameter space because you need "realistically retrievable conditions". Do aerosol models diverge so widely that non-physical or even just different regions of the parameter space would result if aerosol models were utilized with the different formation configurations (even when "scene" conditions were held fixed)? If so, then I guess a correct interpretation is that the imposed constraints listed in Table 2 are more extreme than those that would come from aerosol models. A follow up question would then be: How significant are any of the results in Figures 6-10, for a *stand-alone* aerosol information content study as opposed to a constrained, restricted analysis, of comparative orbital geometry impacts?

I also have some questions about the impact of the results in Figures 6-8. It's mentioned several times throughout the results section that the degrees of freedom of signal gradually increases with number of viewing satellite (for the formation flight) until the 9-satellite configuration and the multi-angle satellite with 9 view angles have nearly indistinguishable mean values. A similar argument is given for the results of Figure 7 and Figure 8, where here a reduction in uncertainty in AOT and fine mode effective radius is shown with increasing number of satellites in the configuration until mean values achieved match those of a multi-angle satellite sensor. What I'm missing in the discussion is that the vertical spread of the results (indicating the variability due to observation geometry) is often times larger for the flight-formation architectures than the multi-angle single platform satellite. The text focuses on the mean values, but does not the larger variability due to observation geometry suggest greater potential for larger uncertainties in global aerosol properties and also that the "time to detect" a trend in aerosol properties would increase? For example, if aerosol models are built using accumulated statistics from global aerosol retrievals, it would seem possible that these models become more uncertain. It would help the interpretation of these figures if you could provide a way to understand how significant the changes in DFS, AOT, or aerosol fine mode effective radius are. In some cases, the change with formation architecture is rather subtle. Could you, for example, extract an answer from your analysis (or compute a result for one representative case) on how the variability in vertical spread of these results would change if an aerosol model was used to constrain the state space as opposed to the imposed constraints in Table 2? Would the vertical variability change (increase) and by such a degree that the changes due to formation architecture differences become non-important?

In this section, I've identified some concerns with the information content analysis. However, I admit that for "relative comparisons", which is the stated goal of this study, much of the fallout from these concerns is diminished because the analysis as described was performed in the same way to every orbital architecture, whether a formation of small satellites or a multi-angle platform.

Comments on the Conclusion Section

The paper is long and detailed so I think what would be really helpful would be to summarize in the conclusions the aspects of the study that weren't covered.  Additionally,  I identified above some aspects of the results that weren't addressed in the text; these should also be summarized in the conclusion. Also, the conclusions as currently written do not provide a path forward for the next analysis nor identify potential studies for future analysis.

For example:

a)  You state the aerosol remote sensing performed equally well with formation flight small satellites.
-   Were new science capabilities using small satellites identified?
-   Are their potential caveats for this statement (for example, time to detect aerosol property trends if geometry variability induces larger variability in retrieved aerosol products)?

b)  The formation flight satellites did not cover the negative view zenith angles as well. The formation flight satellites had more instances of viewing geometries under direct overhead sun.
-   What are the pros/cons for aerosol remote sensing?

c)  Were uncertainties in pointing considered for the formation flight satellites or multi-angle satellite?
-   How does the cumulative effect of pointing uncertainties compare for the formation flight satellites as opposed to the multi-angle satellite?

d)  The information content analysis used to determine geophysical parameter retrieval capability did not include an aerosol model for constraining the retrieval parameter space. Instead, imposed constraints were placed that were tighter than those that would have been obtained by an aerosol model.
-   Should the readers pay more attention to the absolute values (of DFS, AOT uncertainty, etc.) or the relative change in these values as a function of formation architecture?

e)  I'm missing in the conclusion how you would take this kind of analysis to the next step.
-   Can you list a couple of the many aspects of such observations left to explore.
-   Can they be performed with your info content technique, or is an OSSE required?

Minor comments.
Line 11, pg 1: variety *of* view zenith (missing 'of').
Line 25, pg 10: one to many "other"
Line 33, pg 13: Remer et al., 2006 should be in brackets.

---

## Referee Comment (RC2) · F. Xu (Referee) · 9 May 2018

The paper by Knobelspiesse and Nag performed a systematic information content analysis to evaluate the performance of using multiple small satellites for aerosol and surface (land + ocean) remote sensing. As one of the major advantages, small satellites have the flexibility of multiple path location and being replaceable when necessary. The information content analysis demonstrates that such a flexibility results in a similar accuracy as achieved by setting the same number of view angles with a single instrument on a single platform (such as MISR on Terra). Moreover, it is found that the information content does increase with the increase of number of viewing angles.

This work provides important theoretical support to the design and development of

multi-platform sensors for aerosol remote sensing and is highly appropriate for AMT. I have the following comment for the authors to consider and clarify.

1. Page 10: The authors correctly pointed out three pre-assumptions for applying information content analysis. Should there be another one that "The relationship between measurement errors and retrieval uncertainties are assumed to be linear around the solution" ?

2. Earlier work performed by O. Hasekamp et al. 2010 (theoretical study), L. Wu et al. 2015 (using RSP data), and F. Xu et al. 2017 (using AirMSPI data) used direct retrieval test (alternative to information content analysis) and found a significant gain of of AOT retrieval accuracy when the number of viewing angle increases from 2 to 5 and then a limited gain once the number of viewing angles exceed 5. Though the number of viewing angles in this work starts from 3, it will be helpful to add into simulation the 2-angle case and compare these earlier work the retrieval uncertainty as a function of view angles (such as Fig. 7, but plotting absolute AOT error), and comment the difference if there is. This may help the readers be aware of the errors caused by using different analysis approach.

3. Do I understand correctly that the authors conclude the specific location of viewing angle (or "observation geometry" as in the paper) has very limited impact on aerosol/surface retrieval accuracy as long as their number are same ? If so, I'm confused. For a certain number of viewing angles, the spread of the degree of freedom (DOF) in Fig 6 spans a range that can cover the difference in mean DOF caused by varying 4-5 view angles. This is indeed a huge impact. Please clarify.

4. Page 19, paragraph 2, it is not easy for readers to capture these remarks from Figure 6. It is better to add another plot showing the delta_DOF as a function of number of viewing angles. Moreover, I see from the bottom right panel of Fig 6 a gradual increase of DOF from using 5-6 angles, 6-7 angles, and then 7-8 angles. And convergence seems not achieved by use of 9 angles. I suggest the authors setting more angles for

test and plot delta_DOF to justify the convergence. Even claimed as "seven or eight satellite configuration" are capable enough, it is different than earlier finding that AOT retrieval accuracy gain converges at five angles. This needs some comments.

5. The authors correctly uses the chain rule to calculate the AOT uncertainty. To be more complete, please describe more explicitly after Eq.(2) that the square root of the diagonal term of S matrix represents the uncertainty of the retrieval parameters. This would be better than describing it in the figure caption.

6. P23, Section 4,5, the authors are trying to use the off-diagonal terms of the retrieval error correlation matrix (Eq.6) to analyze the cross-contamination between different retrieval parameters. It is stated that "large off diagonal values indicate a smaller volume in retrieval State space, an indication of higher information content for that pair of parameters." Please be more explicit about the physical interpretation behind the relation between diagonal and off-diagonal terms. For example, does the author mean in contrast to the diagonal term, large off-diagonal term means retrieval error of the two quantities are less correlated and therefore easy to decouple ?

7. In addition to the DOF and retrieval uncertainty analysis for the AOT (e.g. Figs 6-7), could the authors add a similar analysis for aerosol single scattering albedo (and maybe an extra case for smoke or dust aerosols) ? This will help the readers understand the role of using polarization in constraining aerosol single scattering albedo retrieval.

---

## Author Comment (AC2) · 29 May 2018

Reviewer response for AMT-2017-473 "Remote sensing of aerosols with small satellites in formation flight" Kirk Knobelspiesse and Sreeja Nag.

**Response to reviewer #2**

The paper by Knobelspiesse and Nag performed a systematic information content analysis to evaluate the performance of using multiple small satellites for aerosol and surface (land + ocean) remote sensing. As one of the major advantages, small satellites have the flexibility of multiple path location and being replaceable when necessary. The information content analysis demonstrates that such a flexibility results in a similar accuracy as achieved by setting the same number of view angles with a single instrument on a single platform (such as MISR on Terra). Moreover, it is found that the information content does increase with the increase of number of viewing angles.

This work provides important theoretical support to the design and development of multi-platform sensors for aerosol remote sensing and is highly appropriate for AMT. I have the following comment for the authors to consider and clarify.

We are grateful for the reviewer's helpful and constructive thoughts on our manuscript.

1. Page 10: The authors correctly pointed out three pre-assumptions for applying information content analysis. Should there be another one that "The relationship between measurement errors and retrieval uncertainties are assumed to be linear around the solution"?

Yes, and we added this to the document. This was mentioned it elsewhere in the paper as a uniform assumption of the error propagation technique proposed by Rodgers, but it is good to also mention it again.

2. Earlier work performed by O. Hasekamp et al. 2010 (theoretical study), L. Wu et al. 2015 (using RSP data), and F. Xu et al. 2017 (using AirMSPI data) used direct retrieval test (alternative to information content analysis) and found a significant gain of AOT retrieval accuracy when the number of viewing angle increases from 2 to 5 and then a limited gain once the number of viewing angles exceed 5. Though the number of viewing angles in this work starts from 3, it will be helpful to add into simulation the 2-angle case and compare these earlier work the retrieval uncertainty as a function of view angles (such as Fig. 7, but plotting absolute AOT error), and comment the difference if there is. This may help the readers be aware of the errors caused by using different analysis approach.

Although we plotted the analysis in figure 7 in terms of relative uncertainty, one can observe a decrease in uncertainty consistent with the work you quote, that tapers off around 5 viewing angles. This is most obvious for the lower simulated AOT's, which is part of our motivation for plotting relative AOT uncertainty. The ability to retrieve aerosol optical properties is directly related to AOT... in other words, better aerosol retrievals if there are more aerosols. It would be nice to add 2 viewing angles to this study, but that would involve starting over from the beginning in terms of orbit design and information content, and wouldn't modify an assessment of our core hypothesis, to compare 9 angle views on a single satellite vs nine single angle satellites.

3. Do I understand correctly that the authors conclude the specific location of viewing angle (or "observation geometry" as in the paper) has very limited impact on aerosol/surface retrieval accuracy as long as their number are same ? If so, I'm confused. For a certain number of viewing angles, the spread of the degree of freedom (DOF) in Fig 6 spans a range that can cover the difference in mean DOF caused by varying 4-5 view angles. This is indeed a huge impact. Please clarify.

Yes, specific viewing geometries have highly variable information content, and you correctly note the evidence for this in the range of DFS in fig. 6. However, satellites flying in formation do not maintain a specific measurement geometry, rather, this varies throughout the orbit. So, in aggregate over the entirety of an orbit, the number of

viewing angles defines the information content of a scene (assuming they are well dispersed within the observing geometry, as is the case for the orbit simulations we used).

4. Page 19, paragraph 2, it is not easy for readers to capture these remarks from Figure 6. It is better to add another plot showing the delta\_DOF as a function of number of viewing angles. Moreover, I see from the bottom right panel of Fig 6 a gradual increase of DOF from using 5-6 angles, 6-7 angles, and then 7-8 angles. And convergence seems not achieved by use of 9 angles. I suggest the authors setting more angles for test and plot delta\_DOF to justify the convergence. Even claimed as "seven or eight satellite configuration" are capable enough, it is different than earlier finding that AOT retrieval accuracy gain converges at five angles. This needs some comments.

We revised our statement "For scenes over the ocean, in fact, the DF S tends to level off after five or six satellites. This would indicate that only that many view angles are required, at least as expressed by the DFS" to "For reflectance-only scenes over the ocean, in fact, the DFS tends to level off after five or six satellites, indicating diminishing returns with more angles. Polarimetric ocean, and both reflectance-only and polarimetric land scenes benefit from additional viewing angles, although the DFS increase becomes more gradual."

Ultimately, we address our primary hypothesis by comparing the 9 view / 1 satellite to the 1 satellite / 9 view case, which shows equivalent DFS (and parameter uncertainties) for the two systems.

5. The authors correctly uses the chain rule to calculate the AOT uncertainty. To be more complete, please describe more explicitly after Eq.(2) that the square root of the diagonal term of S matrix represents the uncertainty of the retrieval parameters. This would be better than describing it in the figure caption.

The sentence following equation 2 is: "The diagonals of this square matrix correspond to squared uncertainties associated with each parameter in x, while off diagonal elements are the covariances between them."

6. P23, Section 4,5, the authors are trying to use the off-diagonal terms of the retrieval error correlation matrix (Eq.6) to analyze the cross-contamination between different retrieval parameters. It is stated that "large off diagonal values indicate a smaller volume in retrieval State space, an indication of higher information content for that pair of parameters." Please be more explicit about the physical interpretation behind the relation between diagonal and off-diagonal terms. For example, does the author mean in contrast to the diagonal term, large off-diagonal term means retrieval error of the two quantities are less correlated and therefore easy to decouple ?

This is a subtle issue that we've attempted to describe in more detail in this version of the manuscript. We use the example of the anti-correlation present in the AOT and effective radius (for the same size mode). Physically, these parameters should be uncorrelated, (one is extrinsic, the other intrinsic). Here's what we put: "All scenes demonstrate a strong anti-correlation between AOT and the effective radius of the same size mode. Physically, these parameters should be uncorrelated, since effective radius is an intrinsic optical parameter, while AOT is an extrinsic parameter expressing total column extinction. Thus, our assumption of no correlation between these parameters in the a priori error covariance matrix, S\_a in Equation 2 is probably valid. This would thus indicate that the source of the anti-correlation is the nature of parameter space expressed in the Jacobians. In practice it would not indicate a relationship in the retrievals of the parameters, but that if one retrieved parameter were wrong, we could expect the other parameter to also be wrong (but in the direction of the opposite sign)."

7. In addition to the DOF and retrieval uncertainty analysis for the AOT (e.g. Figs 6 - 7), could the authors add a similar analysis for aerosol single scattering albedo (and maybe an extra case for smoke or dust aerosols) This will help the readers understand the role of using polarization in constraining aerosol single scattering albedo retrieval.

This is probably most appropriate for a subsequent study with a wider scope beyond comparing the 9 view / 1 satellite to the 1 satellite / 9 view case. That said, the fine mode imaginary refractive index (related to SSA) was a free parameter for the land scenes, although Figure 9 shows low averaging kernel values for that parameter (indicating low sensitivity). Although we have done so in the past, we wonder if Single Scattering Albedo, bounded between 0 and 1, is an appropriate parameter to examine within the Rodgers style formalism, which requires gaussian error distributions. An alternative metric that might work better could be the absorption optical depth.

---

## Author Comment (AC1)

Reviewer response for AMT-2017-473 "Remote sensing of aerosols with small satellites in formation flight" Kirk Knobelspiesse and Sreeja Nag.

**Response to reviewer #1**

*This article assesses the potential for aerosol remote sensing from a formation flight of small satellites via comparisons with a multi-angle single platform (MISR-like) satellite. The authors present their case that a formation of small satellites, each with a single view angle, could perform as well for aerosol remote sensing as a single platform, multi-angle, satellite with the added bonus that small satellites in formation could be replaced as they age at lower overall cost. They use a combined systems engineering approach and information content analysis to support their conclusions.*
*…*
*In light of the intended impact of the paper, I focus my comments on the orbital geometry findings and the information content assessments. However, it's clear from the text and the results that aerosol remote sensing is very challenging and the retrievals are underdetermined. I admit it's possible that I've missed important and subtle concepts regarding the physics of aerosol remote sensing or misconstrued them.*

We sincerely appreciate the reviewer's thoughtful examination of our manuscript. Aerosol remote sensing is indeed challenging, and with currently available instrumentation, retrievals are fundamentally underdetermined.

*Comments on the Systems Engineering Aspect*

*Here, I focus on the pointing control of the formation flight small satellites because I am not finding a clear relation between the orbital geometry metrics presented in the text and how those uncertainties propagated into the information content analysis. I'm no expert in orbital mechanics, so perhaps I missed it, or did not understand what I read.*

*Attitude vs Position: The authors make the point that the relative positions of the formation flight satellites are less important than their relative attitudes. I believe this makes sense because for aerosol remote sensing, what you are aiming to achieve is a measurement of the spectrum of reflected light at enough view angles – over the same patch of surface area – to observe the BRDF or BPDF function. Thus, the ability of the satellites to be pointed at the same patch of surface area is more important than position control, although the satellites would have to be near enough to each other at a given instance of time to be able to point at the same surface patch, of course. Conversely, a situation where very tight position control of formation flight satellites would be more important could be where the apertures of individual instrument are combined (in processing analysis) to make a "pseudo" large aperture for viewing very distant objects.*

*In light of the above, could you explain how the RAAN and Mean Anomaly uncertainties presented in Table 1 that were propagated in the systems engineering model into root mean square (RMS) differences between predicted BRDF and the "truth" BRDF provide a metric of pointing control? My current interpretation is that the RAAN and Mean anomaly uncertainties are rather supporting the case that the small sats can be controlled in close enough formation to be able to point at the same surface patch, not that they actually did point at the same surface patch. I did note some capabilities listed on page 12 – "adjustments to maintain the formation can guarantee > 80% overlap between ground spots for 0.5 degrees of pointing control and 2 km GPS error." It's this "agility" aspect, and its associated uncertainty of 0.5 degrees of pointing control, that I'm not finding was propagated into the BRDF/BPDF (p6). Was it and how so (and how did the implementation in the analysis differ for the formation flight satellites as opposed to the multi---angle sensor)?*

The RAAN and MAs noted in table 1 are not uncertainties, but actual values of the different satellites with respect to satellite 1 (therefore, sat1 is 0/0 for all formations). The reviewer is correct in noting that "*the RAAN and Mean anomaly  are rather supporting the case that the small sats can be controlled in close enough formation to be able to point at the same surface patch*". The given values of RAAN/MA have been found to be appropriate to control the attitude of the formation such that they *can* point at the same surface patch,

assuming "*0.5 degrees of pointing control and 2 km GPS error*", which is very reasonable by commercial small satellite standards. Since BRDF or BPDF computed in the science evaluation or information assessment model is defined for an infinitesimally small point on the surface, we have not propagated errors due to non-perfect overlap of the satellite swaths as a result of pointing inaccuracies.

Nag 2016b shows that >80% overlaps are achievable for all formation architectures considered, from a hardware modeling standpoint. Nag 2018 shows that this "agility" is possible, from the attitude control system scheduling and autonomous software standpoint. The multi-sensor satellite (used as a comparison to the formation satellites) needs no agility because its sensors are fixed to the satellite at defined angles in the forward and aft directions. The fixed ground tracks of the fixed sensors eliminated the need for studying the overlap of their swaths because it is expected to be nearly 100%. To clarify the above, the following line in the paper "The orbital elements proposed above are achievable within commercial small satellite technology" has been replaced with the following:

"The orbital elements proposed above are achievable within commercial small satellite technology. These elements allow for relative separation between the formation's satellites, such that they can point at the same surface spot near-simultaneously. Nag et al. (2018) has confirmed software algorithms that allows for autonomously scheduling the attitude control of multi-satellite systems, to make coordinated observations for customized applications such as the multi-angular we propose."
Nag et al. (2018) => S. Nag, A.S. Li, J.H. Merrick, "Scheduling Algorithms for Rapid Imaging using Agile Cubesat Constellations", COSPAR Advances in Space Research 61, Issue 3 (2018), 891-913.

Finally, to respond to the question: "*Are the capabilities listed in the paper based on measurement requirements for aerosol remote sensing?*" The answer is yes, in a very general sense. We choose an amalgam of existing or planned spacecraft as the basis for our study. MISR, obviously, is the basis for the multi-angle views, and upcoming instruments such as MAIA and 3MI are the basis for the polarization sensitivity and SWIR channel.

*Observing geometries: I observe some characteristics in Figure 2 that were not discussed in the paper. Firstly, when I compare the simulated geometries of the formation flight satellites (left plot) with the multi---angle satellite (right plot), I notice that the formation flight satellites did not cover the negative view zenith angles beyond approximately ---60 degrees. Does this meet BRDF measurement requirements for aerosol remote sensing (referred to on pg 5), and what are these measurement requirements (list in paper or provide citation)? In the subsequent analysis, how does that absence of observations factor into the RMS differences in observed (simulated) BRDF relative to the "truth" airborne CAR measurements?*

By negative view zenith, we assume the reviewer means zeniths > 60 deg around the 180 deg azimuth line, and they are correct in noting sparser measurements by the formations in that region of the angular plane. The sparsity and generally the non-uniformity of measurements is due to the fact that the formation geometry with respect to the Earth or the Sun is not constant. However, the proposed formations in Table 1 and whose measurements are plotted in Figure 2, are those that not only meet the measurement requirements but also simulate the minimum BRDF RMS error relative to CAR measurements. The following sentence has been added to the end of Section 2.1: "Nag et al 2016a and 2016b detail the measurement requirements for surface BRDF observations, and the process of computing RMS errors with respect to reference data (e.g. CAR) so as to optimize formation designs that minimize errors. Nag et al (2017b) demonstrates that commercial payloads flown at the altitude and angular ranges proposed, are capable of meeting the above measurement requirements."
Nag et al (2017b) => S. Nag , T. Hewagama, G. Georgiev, B. Pasquale, S. Aslam, C. K. Gatebe, "Multispectral Snapshot Imagers onboard Small Satellite Formations for Multi-Angular Remote Sensing", IEEE Sensors Journal 17, no. 16 (2017), 5252-5268.

*Secondly, I also notice that the flight formation architectures have viewing geometries under overhead sun conditions (SZA = 0), whereas the multi---angle satellite configuration did not. Are there benefits for aerosol remote sensing that you can identify that would be aided by overhead sun angles? I could imagine that there may be benefits and limitations. For example, measured signals would be larger, but perhaps the surface component of that signal becomes even more dominant than the aerosol component?*

The overhead sun conditions were found to benefit surface BRDF RMS errors, therefore formation architectures that sampled the slope of the hotspot and glint areas of the angular plane performed better than others. From the aerosol remote sensing standpoint - the impact of higher overhead sun angles is expressed in the results of the information content study, which show that whatever potential improvements may exist for overhead sun angles do not result in a global improvement of the single angle formation instruments compared to multi-angle instruments.

*Finally, because the study's strength is in relative comparisons between formation flight architectures and a multi---angle sensor, it's fair to ask if these differences were incorporated into the subsequent information content assessments, or if the analysis was performed such that the set of viewing geometries and solar zenith angles were first pre---filtered to those common to both small satellites and multi---angle sensor prior to doing the information content assessments. The text, however, does make clear that spectral sensitivity and measurement uncertainty assumptions have been kept the same for both small satellites and multi---angle sensor measurement error assumptions.*

These differences are indeed incorporated into the information content assessment, as the viewing geometry is quite important for aerosol retrievals.

*Regarding Figure 3, I don't understand the difference between "science performance" and "science error for engineering design"?*

"Science error for engineering design" is just the science performance that is iteratively used to redesign the systems engineering such that the performance is within acceptable bounds. We have modified the figure to be less confusing to the reader.

*Also, it's a small thing, but what is the yellow symbol below the BRDF plot –is it the sun?*

Yes, the yellow blob is the Sun. However, the figure inside the science evaluation model box on the right should be figure 4. We have corrected figure 3 to reflect so now.

*Comments on the Information Content Analysis*

*The authors utilized a statistical technique derived from general inverse theory commonly called optimal estimation [Rodgers, 2000] in their analysis. … The forward model, representing aerosol physics as best known, provides the mapping between measurement space 'y' and parameter space 'x'- for the constructed sensitivities (Jacobians) of the measurements to the parameters.*

*It's too simplistic to represent the forward model as 'F(x) = y' [line 4, pg 7] and in Figure 4. The forward model will also require some assumed model inputs, some (all?) of which you list in Table 2 (the non-bolded values) that will have their own set of uncertainties. Let's call the necessary, but assumed, model inputs 'b'. So, the mapping from observation space, 'y', to parameter space 'x' will also rely on the accuracy of these 'b' values (line 8-9, p7), and how well natural variability in these 'b' values is captured.*

We were aiming for simplicity in the description of our technique, but you are correct that we oversimplified. In light of this and other comments later on, we have modified this section to indicate (and describe) F(x,b)=y, and also modified Figure 4 accordingly. This will also help us reinforce the point of conducting relative information content tests, in which the sensitivity to b is minimized.

Corrected description is now: "Connecting the two is the forward (RT) model, (F(x,b)=y), which produces a simulated observation, y, given geophysical parameters, x, and other required model inputs, b. The difference between x and b, for the purposes of remote sensing, is that the former are the parameters one wishes to retrieve, while the latter are required to simulate an observation (such as total atmospheric pressure), but are either

parameterized or specified with ancillary data. As we shall see later, we have structured our IC analysis to minimize the impact of changes in b for different systems."

*I'm missing discussion of the physics of aerosol retrievals and what parameters are retrieved as opposed to what parameters are derived from the retrievals. I initially read Table 2 such that bolded variables were the retrieved parameters. However, discussion of AOT (centered around Equation 4) suggests that AOT is actually a derived parameter.*

We expanded the fifth paragraph of the introduction in order to provide a more physical description of the retrieval process: "…Such models compute a BRDF, which is sampled to simulate measurement vector y' given a set of descriptive geophysical scene parameters, x, and other ancillary information b. y and y' are then compared, and x iteratively adjusted (by a variety of methods) until the closest match can be found. The ability to successfully converge to a solution depends on measurement system characteristics, RT model fidelity, and other factors. In this study, we are concerned with the impact that measurement characteristics, specifically observation geometry, have on the ability to accurately determine x. These parameters are indicated in bold in Table 2, whereas other parameters held fixed in the analysis can be considered components of b. The limited information content of the system drives the partition between x and b in our analysis, and is why we choose to compare information content relative to a single multi-angle instrument baseline. The b applied in all cases is the same. In practice, real retrievals may require the use of aerosol models, which address limited information content by constraining parameter space."

Although we don't mention AOT around equation 4, perhaps the confusion is that the AOT is retrieved independently for each size mode, whereas the total AOT (= AOT_f + AOT_c) is derived using equation 4.

*Then, regarding Equation 4, while I do understand that it has been applied in other studies under presumptions that variables can be differentiated I feel I should point out that it's a weakness in the information content analysis. I'm not trying to lessen your efforts or look for a citation, but I've recently begun grappling with those kinds of challenges and made some advances in quantifying the information that is shared between physical variables (cloud optical thickness and particle size in cloud remote sensing) that wouldn't need to rely on those presumptions and I wanted to point it out to you for your interest. Here I'm speaking of mutual information content metrics that I think would hold promise for aerosol remote sensing challenges (see for example, Section 6.1 in Coddington et al., [2017], J. Geophys. Res. Atmos., 122, 8079–8100).*

We should indeed cite the GENRA literature as an alternative means of assessing information content, and discuss some of the potential differences. Clearly, GENRA is advantageous in the sense the distribution is not assumed, and its ability to incorporate model errors. However, the far greater state space dimensionality of aerosol retrievals, and their underdetermined nature (compared to cloud retrievals assessed with GENRA) may make explicit error propagation simpler, in a computational sense. This is because the lookup tables utilized in some aerosol retrieval algorithms are often severe constraints on state space required to avoid multiple solutions, so much larger LUT's would be required to use the GENRA technique. It is something we would like to try in the future, however.

Regarding equation 4, we should have noted that it is an approximation such that covariances between pairs of parameters are neglected. The practical reason for this is that those covariances are not well known, and most likely vary within parameter space.

We added to the paragraph following equation 4 to say:
"This presumes that G(x) can be differentiated, which in our example is the case (see Section 3.3). For practical reasons, it also neglects the potential correlation between parameter pairs. This correlation is difficult to characterize, and possibly variable throughout parameter space. An alternative means to address this issue is discussed in Coddington et al. (2017), which examines the information shared between parameters in the context of cloud remote sensing. This builds upon prior work using Generalized Nonlinear Retrieval Analysis (GENRA), as described and applied in Vukicevic et al. (2010); Coddington et al. (2012, 2013). Unlike explicit error propagation

that we use, GENRA calculates the posterior distribution, and thus information content, without the assumption of Gaussian uncertainties. Potential future outcomes of this work is an examination of the practicality of GENRA for the higher dimensional parameter space of aerosol remote sensing."

*The assumption that Sa is diagonal (i.e. no a priori correlation between parameters) (Line 19, pg 17) would seem to be weakly supported, given your results in Figure 10. Is this a common assumption in aerosol information content analysis? The same assumption for Se (measurement uncertainty) seems reasonable.*

Figure 10 shows anti-correlation between the uncertainty of effective radius and optical thickness of the same aerosol size mode. Physically, they should be uncorrelated, since effective radius is an intrinsic optical parameter, while optical thickness is an extrinsic parameter expressing total column extinction. This would indicate that the source of the anti-correlation is the nature of parameter space contained in the Jacobians. As indicated in section 4.5, the retrieval error correlations are not the same as parameter correlations, and large amounts of correlation or anti-correlation indicate a smaller retrieval volume, and thus greater information content. We modified section 4.5 to further elaborate on this issue.

*I'm missing a clear definition of how you use the term "scene" (ex. Line 14, pg 9). Do you mean different surface types, or do you mean a variety of different atmospheric conditions for the same surface? If the latter, would "state space" be a better word choice than "scene"? Based on your decision, you might need to do a word search through the paper to find instances of "scene". Then a follow up question, regarding the iterative computation of Equation 2 for the different scenes: is the a priori error covariance matrix (Sa) held fixed for all scenes? Based on Table 2, I believe that is the case, however, the Jacobian (K) would change.*

We mean the term 'scene' to indicate a particular point in state space and its corresponding location in measurement space. In measurement space, a scene would be a particular measurement vector, y, while in state space it would be the vector of x and b corresponding to y. We added an explicit definition of what we mean by scene following equation 3.

The a priori error covariance matrix is indeed held constant for all the assessments (for which the Jacobian changes). The only justification for doing otherwise would be an algorithm that uses climatology or other knowledge to guide the a priori covariance matrix, a level of complication probably not appropriate for this study. We did indicate in the same paragraph that this is the case.

*It's probably a small thing, but I'm actually not sure what is meant by "perfect algorithm ability to converge to the best retrieval from the observations" (Line 16, pg 10). Is it just that you are referring to a retrieval with multiple local minima and you're looking for the "best" one (where "best" is ideally the correct solution).*

A retrieval algorithm that uses something like an optimal estimation approach may, for example, become trapped in a local minima, or not find the exact true minima because of computational limits on the number of iterations, etc. We are excluding such problems and assuming that a retrieval algorithm is able to find the true minima without error.

*Lines 23-29, pg 15: You are performing an information content assessment without using an aerosol model to constrain the parameter space because you need "realistically retrievable conditions". Do aerosol models diverge so widely that non-physical or even just different regions of the parameter space would result if aerosol models were utilized with the different formation configurations (even when "scene" conditions were held fixed)? If so, then I guess a correct interpretation is that the imposed constraints listed in Table 2 are more extreme than those that would come from aerosol models. A follow up question would then be: How significant are any of the results in Figures 6-10, for a stand-alone aerosol information content study as opposed to a constrained, restricted analysis, of comparative orbital geometry impacts?*

In the case of MISR, the 36 measurements (9 viewing angles X 4 channels) are underdetermined for the retrieval of the dozen or more relevant state space parameters. For retrievals, aerosol models are chosen to exclude

(presumably nonphysical) multiple solutions. However, the constraints provided by these multiple models are difficult to express within our error propagation analysis. Our solution is to instead constrain some of the parameters while letting others vary freely, in an attempt to match the overall amount of constraint imposed by the use of models. While one would not do this for a retrieval algorithm, it is appropriate for an analysis of how uncertainty propagates through the system. Relative assessment of different measurement systems with the same constraints further reduces our sensitivity to these choices.

If we understand your second question properly, our use of orbital geometry impacts is relevant in that the amount of information contained within an aerosol remote sensing measurement varies significantly with observation geometry (see the range of values for a specified AOT in figures 6-8).

*I also have some questions about the impact of the results in Figures 6-8. It's mentioned several times throughout the results section that the degrees of freedom of signal gradually increases with number of viewing satellite (for the formation flight) until the 9-satellite configuration and the multi-angle satellite with 9 view angles have nearly indistinguishable mean values. A similar argument is given for the results of Figure 7 and Figure 8, where here a reduction in uncertainty in AOT and fine mode effective radius is shown with increasing number of satellites in the configuration until mean values achieved match those of a multi-angle satellite sensor. What I'm missing in the discussion is that the vertical spread of the results (indicating the variability due to observation geometry) is often times larger for the flight- formation architectures than the multi-angle single platform satellite. The text focuses on the mean values, but does not the larger variability due to observation geometry suggest greater potential for larger uncertainties in global aerosol properties and also that the "time to detect" a trend in aerosol properties would increase? For example, if aerosol models are built using accumulated statistics from global aerosol retrievals, it would seem possible that these models become more uncertain. It would help the interpretation of these figures if you could provide a way to understand how significant the changes in DFS, AOT, or aerosol fine mode effective radius are. In some cases, the change with formation architecture is rather subtle. Could you, for example, extract an answer from your analysis (or compute a result for one representative case) on how the variability in vertical spread of these results would change if an aerosol model was used to constrain the state space as opposed to the imposed constraints in Table 2? Would the vertical variability change (increase) and by such a degree that the changes due to formation architecture differences become non-important?*

To my eye, the only case where the range of simulated DoF or uncertainty is consistently (for all figures 6-8) wider for the 9view/1satellite vs the 1sat/9view is Reflectance+Polarization, Ocean, and the difference is subtle. The implication is that the *uncertainty* of the parameters has a greater variability throughout the orbit, but *not* the values of those parameters themselves. The mean value is equivalent. So, no, this would not mean a difference in "time to detect" a trend. To address the question regarding significance of the results, DoF is a good overall measure that addresses the core hypothesis of this research (are the different viewing geometries equivalent). For science relevance we would look to the expected uncertainty of individual parameters, (fig 7-8) or the averaging kernel matrix values. In doing so, we need to keep in mind the assumptions regarding constraints on parameter space, etc. addressed above, and the narrow focus this study. It would be very difficult for us to say anything meaningful about the impact on the range of uncertainty values with the use of fixed aerosol models without the means to calculate this. That said, it is difficult to envision why it would be different.

We added a paragraph in section 4.1 to address this issue.

*In this section, I've identified some concerns with the information content analysis. However, I admit that for "relative comparisons", which is the stated goal of this study, much of the fallout from these concerns is diminished because the analysis as described was performed in the same way to every orbital architecture, whether a formation of small satellites or a multi- angle platform.*

Yes, and thank you for your comments. As you note, a core aspect of this work is how the assumptions are uniform for the different platform types.

*Comments on the Conclusion Section*
*The paper is long and detailed so I think what would be really helpful would be to*
*summarize in the conclusions the aspects of the study that weren't covered. Additionally, I identified above some aspects of the results that weren't addressed in the text; these should also be summarized in the conclusion. Also, the conclusions as currently written do not provide a path forward for the next analysis nor identify potential studies for future analysis.*

We added a paragraph to the conclusion further describing next steps and potential future research. Some of the things you mention below are included. There are two categories of next steps. One category requires more information about specific engineering design choices, while the other would be to couple these results with a full OSSE.

*For example:*
*a) You state the aerosol remote sensing performed equally well with formation flight small satellites.*
- *- Were new science capabilities using small satellites identified?*
- *- Are their potential caveats for this statement (for example, time to detect aerosol property trends if geometry variability induces larger variability in retrieved aerosol products)?*

For the first bullet, the answer is no: as we state, there are relatively few differences using small satellites. See above comments regarding second bullet.

*b) The formation flight satellites did not cover the negative view zenith angles as well. The formation flight satellites had more instances of viewing geometries under direct overhead sun.*
*- What are the pros/cons for aerosol remote sensing?*
Like above, pros/cons are small. The systems, at least as far as this IC study, are mostly interchangeable.

*c) Were uncertainties in pointing considered for the formation flight satellites or multi-angle satellite?*
No – address in updated conclusion.
*- How does the cumulative effect of pointing uncertainties compare for the formation flight satellites as opposed to the multi-angle satellite?*
*d) The information content analysis used to determine geophysical parameter retrieval capability did not include an aerosol model for constraining the retrieval parameter space. Instead, imposed constraints were placed that were tighter than those that would have been obtained by an aerosol model.*
*- Should the readers pay more attention to the absolute values (of DFS, AOT uncertainty, etc.) or the relative change in these values as a function of formation architecture?*
The latter – hopefully we've made this clear.

*e) I'm missing in the conclusion how you would take this kind of analysis to the next step. - Can you list a couple of the many aspects of such observations left to explore.*
*- Can they be performed with your info content technique, or is an OSSE required?*
We attempted to address this in our updated conclusion.

*Minor comments.*
*Line 11, pg 1: variety of view zenith (missing 'of').*
*Line 25, pg 10: one to many "other"*
*Line 33, pg 13: Remer et al., 2006 should be in brackets.*

Thanks for these